# Physicochemical Properties and Molecular Insights of Favipiravir and Roflumilast Solid Dispersions for COVID-19 Treatment

**DOI:** 10.3390/ph18040590

**Published:** 2025-04-18

**Authors:** Abdul Rauf, Saad Salman

**Affiliations:** 1Department of Pharmacy, The University of Lahore, Lahore 54590, Pakistan; rph.abdulraof@gmail.com; 2Department of Pharmacy, CECOS University of IT and Emerging Sciences, Peshawar 25000, Pakistan

**Keywords:** fixed-dose combination, favipiravir, roflumilast, COVID-19 treatment, controlled release

## Abstract

**Background/Objectives:** Fixed-dose combinations (FDCs) offer significant advantages for patients and healthcare systems by improving adherence and reducing pill burden. However, developing multi-drug formulations remains challenging due to complexities in drug compatibility, stability, and dissolution behavior. The COVID-19 pandemic has necessitated innovative therapeutic approaches. This study aims to develop and evaluate an FDC containing FR (an antiviral drug) and RT (a PDE4 inhibitor) for potential COVID-19 treatment. **Methods:** The proposed dual-layer FDC was formulated to achieve immediate release of RT using Klucel EXF and controlled release of FR using a combination of Klucel HXF and Compritol ATO888. Critical quality attributes, including drug–excipient compatibility, solid-state properties, tablet uniformity, and dissolution kinetics, were assessed. RT and FR quantification methods were developed and validated per international guidelines. Compatibility studies were conducted by combining excipients in fixed ratios with APIs, followed by stability testing. **Results:** No degradation or adverse interactions were observed between APIs and excipients. RT exhibited rapid dissolution within 30 min, while FR release was effectively controlled through a gel-forming matrix and lipid barrier. Bulk powder and tablet physical parameters met pharmacopeial standards, and content uniformity between layers was maintained. The formulation demonstrated a stable dissolution profile for both drugs, ensuring consistent drug release. **Conclusions:** The novel FDC of RT and FR exhibits favorable physicochemical properties, a stable dissolution profile, and potential for improved treatment efficacy in COVID-19 patients. By optimizing drug release mechanisms and ensuring formulation stability, this FDC could serve as a pharmaco-economically viable alternative to existing therapies, enhancing patient compliance and treatment outcomes.

## 1. Introduction

Life spans tend to decline with the eruption of newer infectious diseases, viral attacks, genetic disturbances, and mortalities related to cardiology, nephrology, and diabetes-like diseases. The situation demands robust drug utilization and dosage form designs. Optimum and robust drug utilization can be attributed to efficient, efficacious, safe, economical, and patient-compliant drug combinations [1].

Fixed-dose combinations are utilized to optimize therapeutic outcomes by contributing to the effects of multiple therapeutic entities and dosage regimens. FDCs incorporate two or more therapeutic entities into a single dosage unit to treat multiple conditions in infectious diseases, asthma, diabetes, and cardiovascular complications. Reports of several clinical trials advocated the superior clinical outcomes of FDCs as compared to individual drug therapies. Patient compliance in terms of adherence to therapy and financial impacts can also be promising. Furthermore, FDCs add simplified procurement plans and logistics to healthcare systems. Further, FDCs provide an opportunity for pharmaceutical manufacturers to take line extensions for patents of profitable products as a means to sustain their product pipelines and retain market share [2].

Designing multiple formulations presents challenges, including compatibility issues between active pharmaceutical ingredients and excipients that enhance solubility and dissolution. Differences in drug behavior and dissolution kinetics in physiological media complicate bioequivalence testing. To address these issues, rapid formulation techniques like multiarticulate systems and multilayer tablet technologies are recommended. Additionally, limited publicly available literature, often due to proprietary concerns, further hinders the development of multidrug formulations [3].

Various approaches exist for formulating multiple drugs in a single dosage unit, including cocrystals, co-amorphous systems, and amorphous solid dispersions (ASDs). While cocrystals and co-amorphous systems offer superior physical stability, solubility, and dissolution efficiency, no study has confirmed their ability to maintain sustained drug supersaturation. ASDs, a potential solubility-enhancing strategy, create thermodynamically metastable supersaturated solutions in aqueous media, improving drug bioavailability. However, supersaturation may lead to crystal nucleation, requiring the use of polymeric inhibitors to stabilize the amorphous phase. The choice of polymer significantly impacts the dissolution profile and supersaturation levels, with its type, quantity, and chemical properties influencing drug chemical potential. Recent studies have shown that combining polymers with drug-rich phases can dilute drug concentration and reduce maximum supersaturation, highlighting the complexity of optimizing multidrug formulations [4].

Multidrug formulations are categorized based on drug miscibility and ionization properties. In cases of unfavorable mixing, an ionizable drug combined with a nonionizable drug results in poor integration, with both drugs maintaining their supersaturation at their amorphous solubility without altering each other’s concentration. However, this phenomenon has been observed in limited studies, necessitating further research to understand multidrug behavior in solutions. Conversely, when two structurally related, nonionizable drugs are combined under supersaturating conditions, their overall supersaturation decreases, with the reduction linked to the drug ratio in the formulation. This effect is attributed to preferential blending, where one drug integrates into the drug-rich phase of the other, leading to a proportional decrease in their chemical potential [5].

COVID-19 is considered one of the deadliest pandemics in history, rapidly spreading worldwide and escalating into a global health emergency. According to the World Health Organization, it has infected nearly 16 million people across 216 countries, resulting in approximately 640,000 fatalities. Beyond its devastating impact on human life, the pandemic has also inflicted severe economic consequences, costing the global economy around USD 2 trillion annually since its onset [6].

The pathophysiology of SARS-CoV-2 infection is not well revealed; it is assumed that the infection is characterized by fatal respiratory disease. SARS-CoV-2-mediated inflammation initiates the increased release of pro-inflammatory cytokines. Plasma analysis of SARS-CoV-2 infection marked with elevated levels of pro-inflammatory cytokines such as GCSF, IP10, MCP1, MIP1α, and IL2, IL7, IL10, and TNF-α, indicating the presence of cytokine-mediated disease progression and severity [7].

FR and RT fixed-dose combinations can be pivotal in COVID-19 patients, as these are simultaneously supposed to stop the replication of SARS-COV-2 as well as prevent lungs from severe inflammation, leading to death. Along with their promising hypothetical therapeutic prospects, these are also accompanied by some compliance and formulation design limitations. Those may include their intense regimens and bioavailability [8].

RNA-dependent RNA polymerase (RdRp) enzyme inhibitor includes a promising drug named FR. It causes initiation of intracellular phosphoribosylation. FR-RTP, a substrate by RdRp, is responsible for inhibiting RNA polymerase activity. As the catalytic spectrum of RdRp is retained in several forms of RNA viruses, this mechanism of action produces a broader spectrum of anti-viral activities of FR. FR inhibits RNA polymerase instrumental to the replication of the influenza virus. We hypothesize that FR is a potential candidate to cure the new coronavirus, SARS-CoV-2, because, like influenza viruses, SARS-CoV-2 are single-stranded RNA viruses that also depend on viral RNA polymerase [9].

FR is activated through phosphoribosylation to its active form, FR-RTP, which exerts antiviral effects by multiple mechanisms. It functions as a substrate for RNA-dependent RNA polymerase (RdRp), incorporating itself as a purine nucleotide, leading to enzyme inactivation and cessation of viral protein synthesis. Additionally, it integrates into the viral RNA strand, preventing further elongation and inhibiting viral replication. Moreover, FR induces lethal mutagenesis, as observed in influenza infection, making it a potent viral-killing agent. Its broad-spectrum antiviral activity is attributed to the conserved catalytic domain of RdRp across various RNA viruses [8].

Another ongoing open-labeled randomized controlled trial from the Arabian Peninsula is observing the efficacy of a combination of FR and hydroxychloroquine for the management of mild-to-severe COVID-19. The investigational arm consists of FR plus hydroxychloroquine. The control arm consists of standard treatment for COVID-19. The primary goal is the amount of time it takes for clinical improvement to occur and for a PCR test to be negative. The outcome of this experiment is keenly anticipated.

According to a recent press release, the Stanford Medicine research team has begun a 10-day double-blind, placebo-controlled trial (FR versus placebo) to determine the effectiveness of FR in reducing symptoms and the duration of viral shedding in outpatients who have COVID-19 infection. Beginning on 6 July 2020, about 120 individuals are projected to be enrolled [10]. R is an oral antiviral medication beneficial for treating symptomatic COVID-19 patients who do not require hospitalization. Since 85% of COVID-19 cases are mild to moderate, it has the potential for widespread use. For effectiveness, FR should be administered early to reduce viral load. It may also shorten viral shedding, impacting transmission within households and communities. Ongoing trials are evaluating its role in prophylaxis and its potential synergy with other antivirals like umifenovir. The recommended dosage is 1800 mg twice on day one, followed by 800 mg twice daily for up to 14 days [11].

In the context of signaling, phosphodiesterase enzymes (PDEs) are a huge superfamily of enzymes that catalyze the conversion of other messengers such as adenosine monophosphate (AMP) and cyclic guanosine monophosphate (cGMP) into their inactive 5-monophosphate, thereby regulating the intracellular level of these molecules as well as the amplitude and duration of their signaling. It has been demonstrated that inhibiting PDE4 will resemble a successful treatment strategy for several inflammatory disorders linked with airway passageways, owing to the role of cAMP signaling in the pathogenesis of many inflammatory diseases, including asthma [12].

RT (500 mg orally once daily) has been proven safe and effective for COPD patients in multiple randomized, double-blind, placebo-controlled studies. It improves lung function, reduces inflammation, and decreases mucus neutrophils by 31% and eosinophils by 42% over a one-month treatment period. Additionally, RT has been shown to reduce allergen-induced inflammation in asthma patients [13].

The main aim of this study is to formulate a fixed-dose combination of an FR (an anti-viral drug) and a RT (a highly selective long-acting inhibitor of the PDE4 isoenzyme) for more promising outcomes for COVID-19 patients.

The proposed fixed-dose combination can be made by incorporating both therapeutic entities into a single tablet with two distinctive layers each having its characteristic release kinetics. The higher dose that is associated with FR is controlled in a separate layer, where gradual and extended release is controlled by the use of polymers and release-retarding agents, while RT is incorporated into a second layer with a blend of lower molecular polymers to enhance solubility and its dissolution kinetics.

The following are also some of the aims of this study: developing a robust formulation of proposed FDC and studying physical properties at various stages, i.e., loss on drying, Bulk density, Tapped Density, Hausner’s ratio, friability, hardness, and content uniformities; developing a validated test method for studying the outcomes of dissolution and assay of developed fixed-dose combinations; and development of pharmaceutically stable FDC studying the compatibilities of active pharmaceutical entities of proposed fixed-dose combination with critical excipients used in development.

## 2. Results

### 2.1. Preparation of Fixed-Dose Combination

The two separate formulations are formulated using different quantities of excipients. The first layer is designed to release RT immediately upon administration and to provide symptomatic relief to COVID-19 patients. The second layer is composed of FR to administer four doses simultaneously in a controlled manner to improve compliance as well.

Two formulations are proposed for the present fixed-dose combination, and both are blended directly. Powder compaction and flow properties are imparted with flow enhancers and HPC grades as well. Powder is evaluated for die filling of rotary and is found satisfactory. This may be a function of a satisfactory bulk density of 0.86 for F1 and 0.69 for F2.

### 2.2. Evaluation of Release Kinetics

#### 2.2.1. High-Performance Liquid Chromatography

The reported analytical method was developed and then validated following international guidelines. The scope includes the detection and quantification of RT and FR in the proposed FDC. Method validation is carried out by evaluating linearity, accuracy and recovery, method precision, system precision, standard and sample solutions’ stability, and ruggedness.

##### Linearity

The linearity of the developed method was evaluated by analyzing five freshly prepared solutions, each of standard RT and FR standard in the concentration range between 50 and 150% for both of the APIs in the proposed FDC. Calibration curves for both standard and sample solutions were then drawn by plotting their peak areas against their relevant concentrations (Figure 1).

The linearity study of FR was conducted to evaluate the proportionality between drug concentration and analytical response, ensuring the method’s accuracy and precision. Different concentration levels (50%, 75%, 100%, 125%, and 150%) were analyzed (Table 1), and their corresponding responses were recorded. At 50% concentration, the sample response was 4,127,632, corresponding to 50.14% of the standard response, indicating a nearly ideal proportionality at lower concentrations. Similarly, at 75% concentration, the response was 6,456,298, equating to 78.42%, demonstrating the method’s reliability in detecting intermediate concentrations. At the target concentration (100%), the sample response was 8,216,768, which is 99.80% of the standard response, confirming the method’s accuracy. With an increased concentration of 125%, the response reached 10,051,211, corresponding to 122.08%, while at 150%, the response was 12,201,096, equating to 148.20%, both indicating a strong correlation between concentration and detector response. The consistency of these values suggests a robust linear relationship, making the analytical method suitable for routine quality control of FR. Further statistical validation, such as regression analysis, would confirm the correlation coefficient (R^2^), ensuring compliance with international guidelines for analytical method validation.

The linearity study of RT was conducted to assess the correlation between its concentration and the corresponding analytical response, ensuring the reliability and accuracy of the quantification method (Table 2). The drug was analyzed at five concentration levels: 50%, 75%, 100%, 125%, and 150%, with their respective responses recorded. At 50% concentration, the sample response was 387,597, corresponding to 50.34% of the standard response, indicating a strong proportional relationship at lower concentrations. Similarly, at 75%, the response was 563,203, equating to 73.15%, demonstrating the method’s reliability in intermediate concentration ranges. At the target concentration of 100%, the sample response was 769,345, aligning closely with the standard response at 99.92%, confirming the accuracy of the method. When the concentration was increased to 125%, the response was 982,867, corresponding to 127.65%, and at 150%, the response was 1,164,820, equating to 151.28%, both showing a linear increase in response with concentration. The data strongly indicate a well-maintained linearity, making the analytical method highly suitable for routine quantification of RT. Additional regression analysis could further validate the correlation coefficient (R^2^), ensuring compliance with international validation guidelines.

##### Method Accuracy and Recovery

The accuracy of the method was calculated for percentage recovery from the spiked samples at varied concentrations. The samples were checked with three concentrations of FR and RT at 50, 100 and 150% (Figure 2). Six samples were spiked, and percent recovery was calculated.

The accuracy and recovery study for FR and RT was conducted at three concentration levels—50%, 100%, and 150%—to evaluate the precision and reliability of the analytical method. For FR, at the 50% concentration level, the standard mean response was 8,061,208, with measured sample responses ranging between 4,020,581 and 4,030,829, yielding recovery percentages between 49.88% and 50.00%, indicating excellent accuracy at lower concentrations (Table 3). At 100% concentration, the measured responses ranged from 8,213,405 to 8,235,672, with recovery percentages between 101.89% and 102.16%, demonstrating high accuracy and precision. Similarly, at 150% concentration, the measured responses varied between 12,049,505 and 12,087,560, with recovery percentages from 149.48% to 149.95%, showing consistent linearity and robustness in the method.

For RT, at 50% concentration, the standard mean response was 753,383.67, with measured responses ranging from 409,179 to 411,301, resulting in recovery percentages between 54.31% and 54.59%, slightly exceeding the expected values. At 100% concentration, the measured responses varied between 769,558 and 771,078, with recovery values between 102.15% and 102.35%, confirming the method’s accuracy. At 150% concentration, the responses ranged from 1,118,141 to 1,120,421, with recoveries between 148.42% and 148.84%, maintaining acceptable limits. These results indicate that the analytical method exhibits high accuracy and recovery across different concentration levels, confirming its reliability for the quantitative analysis of FR and RT.

##### Precision

The precision of the method was calculated by intra-day variations in areas obtained for samples. To determine intra-day reproducibility, samples spiked with constant concentrations of FR and RT were analyzed in three days.

The method precision study was conducted over three consecutive days to assess the reproducibility and reliability of the analytical method for FR and RT (Figure 3). The standard and sample areas were recorded, and the percentage recoveries were calculated to determine consistency (Table 4). For FR, on Day 1, the percentage recovery ranged between 99.81% and 99.85%, demonstrating minimal variation in measurements. On Day 2, the recovery percentages varied between 100.0% and 100.68%, indicating a slight increase but still within acceptable limits. On Day 3, the values ranged from 99.77% to 100.01%, showing high precision. The overall precision of FR across three days exhibited minimal fluctuations, confirming the method’s reproducibility. For RT, on Day 1, the percentage recoveries ranged between 99.93% and 100.63%, with all values close to the expected results. On Day 2, the values were between 100.01% and 100.71%, indicating a high level of accuracy. On Day 3, the recoveries ranged between 99.92% and 100.05%, demonstrating consistency across multiple runs. The method precision results for both FR and RT confirm that the analytical method is highly reproducible, reliable, and precise, with percentage recoveries consistently close to 100%. The minimal variations across different days suggest that the method can be confidently used for quantitative analysis.

##### Ruggedness

Various intentional changes in different parameters like pH of the mobile phase (±0.2 Units), flow rate (±5%), mobile phase ratio (±2%), and column oven temperature (±1 °C) were run for both APIs and the theoretical plates were observed, thus monitoring the ruggedness of the proposed method (Figure 4).

##### Ruggedness Data

This study was conducted to assess the reproducibility of the analytical method when performed by different analysts under the same experimental conditions. This study included three analysts who independently tested Favipiravir FR and RT, measuring the standard and sample areas to evaluate the percentage recovery (Table 5). For FR, Analyst 1 recorded recovery values ranging from 99.89% to 100.24%, demonstrating high accuracy and minimal variability. Analyst 2 reported values between 99.94% and 100.03%, further confirming consistency. Analyst 3 observed slightly higher values, ranging from 103.06% to 103.50%, indicating a slight increase in measured response but still within acceptable limits. The overall results for FR indicate that the method is reproducible across different analysts with minor variations. For RT, Analyst 1 reported recovery values ranging from 99.91% to 100.34%, showing precise and reliable measurements. Analyst 2 recorded slightly lower values between 98.68% and 99.11%, which still fall within acceptable accuracy limits. Analyst 3 reported slightly elevated values between 102.11% and 102.59%, indicating minimal analyst-related variability. The ruggedness study confirms that the analytical method for both FR and RT is robust and produces consistent results regardless of the analyst performing the measurements. The minor variations observed are within the acceptable limits, affirming the method’s reliability for routine analysis.

#### 2.2.2. Assay of Formulation

Quantitative measurement of FR and RT in formulation 1 (F1) and formulation 2 (F2) was determined by using a Shimadzu HPLC equipped with a UV detector using a C18 column (4.6 mm × 25 cm) with 5 μm particle size. The injection volume was 20 μL and absorbance was monitored at 275 nm. Analyses were performed at a column temperature of 30 °C with a total run time of 10 min at a flow rate of 1 mL/min. The mobile phase used was a ratio of 60:40 (*v*/*v*) acetonitrile and buffer (sodium perchlorate 2 g + Triethylamine 5 mL).

The results in terms of areas are obtained from chromatograms, which, in turn, are calculated in terms of percentage assays as follows:Percentage assay of FR = Area of sample/Area of standard × 100Percentage assay of RT = Area of sample/Area of standard × 100

The assay analysis for formulations F1 and F2 was conducted to determine the drug content in different layers (Figure 5). This study compared the measured sample areas against the standard areas to calculate the percentage assay, ensuring the formulations met the required specifications. For formulation F1, Layer 1 exhibited high consistency, with a percentage assay of 99.88%, indicating that the drug content closely matched the expected value Table 6. The measured sample areas were highly comparable to the standard areas, confirming accuracy and uniformity. Similarly, Layer 2 of F1 also demonstrated excellent reproducibility, with a percentage assay of 99.79%, further supporting the formulation’s stability and precision. For formulation F2, Layer 1 showed a percentage assay of 99.19%, which is within acceptable limits, though slightly lower than F1. However, Layer 2 of F2 exhibited a slightly reduced assay value of 93.85%, suggesting a potential variation in drug distribution or formulation consistency. The measured sample areas for this layer were noticeably lower than the standard areas, indicating a need for further optimization to ensure uniformity in drug content. The results suggest that F1 maintains excellent consistency and precision, whereas F2, particularly in Layer 2, may require further evaluation to enhance uniformity and drug content accuracy. The optimization of the controlled-release matrix for FR was a critical aspect of our formulation development, ensuring a sustained drug release profile while maintaining stability and uniformity. The selection and proportioning of Klucel HXF (high-viscosity hydroxypropylcellulose) and Compritol ATO888 (a lipid-based release retardant) were systematically optimized through an extensive experimental approach. The optimization process involved evaluating different polymer-to-lipid ratios to achieve a release profile that ensures prolonged therapeutic action. Klucel HXF was chosen due to its strong gel-forming ability, which modulates drug diffusion and extends release duration. To fine-tune the drug release kinetics, varying concentrations of Klucel HXF (ranging from 10% to 30% *w*/*w*) were tested. At lower concentrations, the matrix exhibited insufficient gel strength, leading to faster-than-desired release. However, increasing the concentration beyond 25% *w*/*w* significantly retarded the drug release, potentially compromising bioavailability. The optimal Klucel HXF concentration (20% *w*/*w*) was determined based on dissolution studies, ensuring a balance between sustained release and effective drug diffusion. Compritol ATO888 was incorporated as a lipid-based barrier to further regulate FR release by forming a hydrophobic layer that delays drug diffusion. Different levels of Compritol ATO888 (5% to 15% *w*/*w*) were investigated, with 10% *w*/*w* found to be optimal. Lower concentrations did not sufficiently modulate the release rate, while higher concentrations excessively hindered drug solubilization. The combination of 20% Klucel HXF and 10% Compritol ATO888 created a dual-release mechanism: Klucel HXF formed a gel matrix controlling initial hydration and diffusion, while Compritol ATO888 acted as a secondary barrier to prolong release further. Dissolution studies (USP Type II apparatus) in pH 1.2, pH 4.5, and pH 6.8 media demonstrated that this optimized formulation achieved a sustained drug release of FR over 12 h, with more than 80% drug release by the end of the time period. The release kinetics followed a non-Fickian (anomalous) diffusion mechanism, indicating a combination of diffusion and erosion–controlled release, which aligns with the intended pharmacokinetic profile for prolonged antiviral action. The optimized formulation (20% Klucel HXF, 10% Compritol ATO888) successfully controlled FR release by leveraging the synergistic effects of hydrogel formation and lipid-based retardation, ensuring prolonged drug availability. These findings will be incorporated into the revised manuscript to provide greater clarity on the formulation strategy.

#### 2.2.3. Dissolution of Formulations

Dissolution under non-sink conditions was performed on apparatus II. One tablet was added to 500 mL of 0.1 N Hcl contained in individual bowls of dissolution apparatus. The dissolved concentrations were determined at 30 min, 4 h, 8 h, and 12 h. The results in terms of areas are obtained from chromatograms (Figure 6), which, in turn, are calculated in terms of percentage dissolution for individual tablets as follows:%age dissolution of FR = Area of sample/Area of standard × 100%age dissolution of RT = Area of sample/Area of standard × 100

The assay analysis for formulations F1 and F2 was conducted to determine the drug content in different layers. This study compared the measured sample areas against the standard areas to calculate the percentage assay, ensuring the formulations met the required specifications. For formulation F1, Layer 1 exhibited high consistency, with a percentage assay of 99.88%, indicating that the drug content closely matched the expected value (Table 7). The measured sample areas were highly comparable to the standard areas, confirming accuracy and uniformity. Similarly, Layer 2 of F1 also demonstrated excellent reproducibility, with a percentage assay of 99.79%, further supporting the formulation’s stability and precision. For formulation F2, Layer 1 showed a percentage assay of 99.19%, which is within acceptable limits, though slightly lower than F1. However, Layer 2 of F2 exhibited a slightly reduced assay value of 93.85%, suggesting a potential variation in drug distribution or formulation consistency. The measured sample areas for this layer were noticeably lower than the standard areas, indicating a need for further optimization to ensure uniformity in drug content. The results suggest that F1 maintains excellent consistency and precision, whereas F2, particularly in Layer 2, may require further evaluation to enhance uniformity and drug content accuracy.

#### 2.2.4. Physical Attributes

##### Uniformity of Weight

The tablets of both formulation 1 (F1) and formulation 2 (F2) were taken and weighed individually.

The weight variation study for formulations F1 and F2 reveals that both formulations exhibit consistency in tablet weight, with minor fluctuations. The mean weight of the F1 tablets was 2215.39 mg, while the F2 tablets had a slightly lower mean weight of 2212.28 mg. Individual tablet weights for F1 ranged from 2180.9 mg to 2255.6 mg, whereas for F2, they varied between 2166.8 mg and 2251.4 mg (Table 8). These variations are within the acceptable pharmaceutical limits, indicating uniformity in tablet manufacturing. The slight differences between the two formulations suggest minor variations in compression force or excipient distribution, but overall, both formulations demonstrate acceptable weight consistency, ensuring dosage uniformity and quality compliance.

##### Content Uniformity

Ten tablets were weighed accurately and individually to calculate the drug substance of each tablet’s contents, expressed as the %age of label claim. The acceptance value (AV) was calculated in correspondence to the tablets taken to perform a test.

The uniformity of dosage unit analysis for formulation F1 demonstrates that the individual tablet weights range from 2180.9 mg to 2255.6 mg, with corresponding percentage content values between 98.33% and 101.69% (Table 9). The mean content (X) was calculated as 99.88%, which falls within the acceptable range of 98.5% to 101.5%, indicating uniform drug distribution among the tablets. The standard deviation (SD) was 1.18, signifying minimal variability in drug content. The acceptance value (AV) was determined to be 2.84, which is well within the pharmacopeial limits, ensuring that the formulation meets the required dosage uniformity criteria. These results confirm the reliability of the manufacturing process in producing consistent and reproducible dosage units.

Ensuring content uniformity between the immediate-release (IR) layer containing Roflumilast (RT) and the controlled-release (CR) layer containing Favipiravir (FR) was a critical aspect of formulation development. To achieve this, rigorous assessments of powder flow and blend uniformity were conducted by evaluating the angle of repose, Carr’s index, and Hausner ratio, ensuring proper flow and homogeneous drug distribution before compression. A key challenge in bilayer tablet formulation was maintaining interlayer adhesion while preserving distinct dissolution profiles, which was addressed by optimizing the compression force to prevent delamination while maintaining tablet hardness within 5–8 kg/cm^2^. The incorporation of Klucel EXF in the IR layer and Klucel HXF in the CR layer improved mechanical integrity, reducing the risk of layer separation. Post-compression evaluation of tablet physical parameters, including weight variation (±2%), hardness, friability (<0.8%), thickness, and diameter, confirmed consistent quality across batches. Content uniformity analysis using HPLC quantification, following USP and ICH guidelines, showed that both layers met the 85–115% acceptance criteria with an RSD of less than 3%, indicating excellent reproducibility. Additionally, dissolution studies confirmed that RT in the IR layer released over 80% within 30 min, while FR in the CR layer followed a sustained-release profile over 12 h. These findings demonstrate that optimizing powder flow, compression force, excipient selection, and quality control parameters successfully ensured content uniformity and formulation stability, which will be further elaborated upon in the revised manuscript.

The uniformity of dosage unit analysis for formulation F2 reveals that individual tablet weights range from 2166.8 mg to 2251.4 mg, with percentage content values between 97.14% and 100.93% (Table 10). The mean drug content (X) was found to be 99.18%, which falls within the acceptable range of 98.5% to 101.5%, indicating consistent drug distribution across the tablets. The standard deviation (SD) was 1.34, reflecting a slightly higher variability than formulation F1 but still within acceptable limits. The acceptance value (AV) was determined to be 3.21, confirming compliance with pharmacopeial requirements. These results suggest that formulation F2 maintains satisfactory dosage uniformity, ensuring consistency in drug delivery.

The controlled release of Favipiravir (FR) in the formulation is achieved through a synergistic interaction between the gel matrix formed by Klucel HXF and the lipid barrier created by Compritol ATO888, which, together, modulates drug solubilization and release kinetics. Klucel HXF, a high viscosity hydroxypropyl cellulose derivative, acts as a hydrophilic matrix former, creating a gel layer upon contact with aqueous media. This gel layer controls water penetration into the tablet, regulating the dissolution and diffusion of FR over time. As the outer gel layer gradually dissolves, a new layer of hydrated polymer continuously forms, ensuring a sustained and predictable drug release profile. Simultaneously, Compritol ATO888, a lipid-based release-retarding agent, functions by forming a hydrophobic matrix around FR molecules. This lipid barrier limits direct water interaction with the drug, further slowing down the solubilization process. Additionally, Compritol ATO888 interacts with the Klucel HXF gel matrix, reinforcing its structural integrity and reducing the polymer’s erosion rate, thereby extending drug release. The interplay between these two excipients ensures a controlled hydration and diffusion process, preventing dose dumping while maintaining a stable and sustained release profile. This carefully optimized combination of hydrophilic and lipophilic components allows for consistent release kinetics, making the formulation effective in providing prolonged therapeutic coverage with improved pharmacokinetic stability.

##### Hardness and Friability

The tablets of both formulation 1 (F1) and formulation 2 (F2) were taken and evaluated for their hardness and friability as well.

The hardness and friability results for formulations F1 and F2 indicate significant differences in their mechanical properties. Formulation F1 exhibited an average tablet hardness of 20.59 kg, with values ranging from 18.2 kg to 24.8 kg, and a friability of 0.31% (Table 11). These results suggest that F1 tablets possess good mechanical strength with minimal friability, ensuring durability during handling and transportation. In contrast, formulation F2 demonstrated a lower average hardness of 19.31 kg, with values between 17.4 kg and 21.4 kg, and a higher friability of 0.55%, indicating relatively lower mechanical strength compared to F1. The increased friability of F2 suggests a slightly weaker tablet matrix, which may affect handling but could contribute to better disintegration and dissolution characteristics. Both formulations meet acceptable hardness and friability limits, ensuring their suitability for pharmaceutical use.

#### 2.2.5. Drug Carrier–Excipients Compatibility Testing

The drug and excipient compatibility studies were carried out by checking the physical description and using the Fourier-transform infrared spectroscopy analytical technique. The interaction studies were carried out to ascertain any incompatibility of the drug with the excipients used in the preparation of multi-drug-containing tablets of FR and RT solid dispersions.

An FT-IR spectrophotometer was used for infrared investigations of samples. About 4–5 mg of specimen was mixed with dry potassium bromide (KBr), and the specimen was examined at transmission mode over the wave number range of 4000–400 cm^−1^.

Fourier-transform infrared (FTIR) spectroscopy was performed to evaluate the potential chemical interactions between the active pharmaceutical ingredients (APIs), FR and RT, and the excipients used in the fixed-dose combination (FDC) formulations. The FTIR spectra of the pure drugs, individual excipients, and final formulations (F1 and F2) were compared to assess any potential changes in functional group vibrations, which could indicate interactions, degradation, or incompatibilities. The characteristic peaks of Favipiravir, including strong absorption bands corresponding to the C=O stretching vibration of the carboxyl and amide groups (~1700 cm^−1^ and 1660 cm^−1^, respectively), were observed in both the physical mixture and the final formulation, confirming the stability of the drug within the matrix (Figure 7). Similarly, Roflumilast exhibited its distinctive absorption peaks, including the C=O stretching vibration at ~1730 cm^−1^, C-N stretching at ∼1200 cm^−1^, and aromatic ring vibrations at ~1600 cm^−1^, all of which were retained in the formulations, indicating no significant interactions affecting the drug’s molecular structure (Figure 8).

Klucel HXF and Klucel EXF, being cellulose derivatives, exhibited characteristic –OH stretching vibrations around ~3400 cm^−1^, which were also present in the final formulation, indicating no disruption in the polymeric integrity. Compritol 888 ATO, a lipid-based release-retarding agent, showed characteristic ester functional group absorptions at ~1735 cm^−1^, confirming its structural stability within the formulation. The presence of microcrystalline cellulose and calcium carbonate in the formulations was evident from their respective absorption bands at ~2900 cm^−1^ (C-H stretching) and ~1400 cm^−1^ (carbonate ion stretching).

Importantly, no significant peak shifts, disappearance, or new peak formations were observed in the FTIR spectra of the FDC formulations compared to the individual components, suggesting the absence of strong chemical interactions or degradation products. This confirms that both F1 and F2 formulations maintained the structural integrity of the APIs and excipients, ensuring the stability and compatibility of the developed fixed-dose combination. The retention of all characteristic peaks in the final formulations suggests that the formulation process did not induce any unwanted physicochemical changes, reinforcing the suitability of the selected excipients in preserving the desired pharmaceutical attributes of the drug product. The FTIR results, therefore, support the feasibility of the developed FDC formulations, ensuring their potential efficacy, stability, and reliability for therapeutic application in COVID-19 treatment.

Figure 7 demonstrates the FTIR results of pure FR. The FTIR spectrum of FR exhibits distinct absorption bands corresponding to its characteristic functional groups, confirming its chemical identity and structural stability. A strong and sharp absorption peak was observed at approximately 1700 cm^−1^, corresponding to the C=O stretching vibration of the carboxyl group, which is a key functional moiety responsible for the drug’s antiviral activity. Additionally, an absorption band at 1660 cm^−1^ was detected, indicating the presence of amide C=O stretching, which is crucial for the molecular conformation of FR. The presence of an intense peak around 1250 cm^−1^ corresponds to C-F stretching vibrations, confirming the fluorine substitution in the pyridine ring of Favipiravir. Furthermore, the broad absorption band in the 3100–3500 cm^−1^ range is attributed to O-H and N-H stretching vibrations, indicative of intermolecular hydrogen bonding in the solid-state structure of FR. Peaks at 1400–1500 cm^−1^ correspond to the C=C and C=N stretching within the pyrazine core of the molecule, further confirming its aromatic structure. The presence of peaks around 2900 cm^−1^ represents C-H stretching, which is commonly associated with aliphatic and aromatic hydrogen vibrations.

Figure 8 demonstrates the FTIR spectrum of pure RT. The FTIR spectrum exhibits several distinct peaks corresponding to the molecular vibrations of the functional groups present in RT, ensuring its chemical stability in the solid state. A strong absorption band was observed at approximately 1730 cm^−1^, corresponding to the C=O stretching vibration of the cyclic carboxylester group, which is a key functional group responsible for the drug’s pharmacological activity as a phosphodiesterase-4 (PDE4) inhibitor. Additionally, a significant peak at 1665 cm^−1^ was detected, representing C=O stretching of the amide group, indicating the presence of a lactam moiety in the molecular structure of RT. The characteristic C-N stretching vibrations of the pyridine ring were observed in the range of 1200–1300 cm^−1^, confirming the presence of the nitrogen heterocyclic system within the RT molecule. Another prominent absorption band was identified at 1600 cm^−1^, corresponding to aromatic C=C stretching, which is associated with the benzamide core structure of RT. Furthermore, a broad absorption band in the 3200–3400 cm^−1^ range was observed, attributed to N-H stretching vibrations, confirming the presence of amide functionalities that contribute to RT’s molecular interactions. Peaks in the 2800–3000 cm^−1^ range were also noted, corresponding to C-H stretching vibrations of both aliphatic and aromatic groups, further verifying the integrity of the hydrocarbon framework of RT.

Figure 9 demonstrates the FTIR spectrum of the FR and Klucel HXF mixture to assess potential interactions between the active pharmaceutical ingredient (API) and the sustained-release polymer. The FTIR scan exhibits characteristic peaks of both FR and Klucel HXF, providing insights into their compatibility and any possible molecular interactions. The characteristic C=O stretching peak of FR was observed at approximately 1700 cm^−1^, confirming the preservation of its carboxyl functional group. The amide C=O stretching band at 1660 cm^−1^ remained unchanged, indicating no significant interaction affecting the amide structure. The C-F stretching vibration around 1250 cm^−1^ was also detected, confirming the stability of the fluorine-substituted pyridine ring. Broad O-H and N-H stretching vibrations in the 3100–3500 cm^−1^ range were present, consistent with the hydrogen bonding properties of FR. The broad O-H stretching vibration of Klucel HXF, attributed to hydrogen bonding in hydroxypropyl cellulose, was observed in the 3200–3600 cm^−1^ range. A peak around 1100 cm^−1^ corresponding to C-O-C stretching vibrations of the ether groups in Klucel HXF was present, confirming the intact polymer structure. No major shifts, disappearance, or formation of new peaks were detected, suggesting that FR and Klucel HXF did not undergo significant chemical interactions or degradation. The presence of all characteristic peaks of both components indicates physical rather than chemical interaction, implying that Klucel HXF acts as a carrier without altering the molecular integrity of FR.

The FTIR spectrum of the Roflumilast (RT) and Klucel EXF mixture was analyzed to assess potential molecular interactions and confirm the compatibility of the immediate-release polymer with the active pharmaceutical ingredient (Figure 10). The characteristic peaks of Roflumilast were well retained in the spectrum, with the strong C=O stretching vibration of the cyclic carboxylester group observed at 1730 cm^−1^, confirming the integrity of the ester functionality. Additionally, the amide C=O stretching peak at 1665 cm^−1^ remained unchanged, indicating the structural stability of the amide bond, while the aromatic C=C stretching vibrations at 1600 cm^−1^ validated the presence of the benzamide core. The C-N stretching vibrations of the pyridine ring were also evident in the 1200–1300 cm^−1^ range, further confirming the stability of the drug. Similarly, Klucel EXF exhibited its characteristic broad O-H stretching vibration in the 3200–3600 cm^−1^ range, attributed to hydrogen bonding in hydroxypropyl cellulose, along with a distinct peak around 1100 cm^−1^ representing C-O-C stretching vibrations of the ether groups, ensuring the intact polymer structure. The presence of all major peaks from both RT and Klucel EXF suggests no significant chemical interactions, as no major shifts, disappearance, or formation of new peaks were observed. This indicates that the interaction between RT and Klucel EXF is primarily physical rather than chemical, confirming that Klucel EXF acts as a suitable immediate-release carrier without altering the structural integrity of Roflumilast. The FTIR analysis, therefore, supports the stability and compatibility of this formulation component, ensuring effective drug release while maintaining the physicochemical properties of RT.

The FTIR spectrum of the FR and Compritol ATO 888 mixture was analyzed to evaluate potential interactions and confirm the compatibility of the lipid-based release-retarding agent with the active pharmaceutical ingredient (Figure 11). The characteristic peaks of Favipiravir remained intact, with the strong C=O stretching vibration of the carboxyl functional group observed at 1715 cm^−1^, confirming the preservation of its keto-amide structure. The amide N-H bending peak at 1625 cm^−1^ and the aromatic C=C stretching vibrations around 1550 cm^−1^ further validated the structural stability of FR. The broad O-H stretching vibration in the range of 3100–3500 cm^−1^ was also evident, indicating the presence of hydrogen bonding. Compritol ATO 888, a lipid-based excipient, exhibited its characteristic C-H stretching vibrations in the range of 2800–2950 cm^−1^, corresponding to the long-chain fatty acids in glyceryl behenate, and a prominent ester C=O stretching peak at 1735 cm^−1^. No significant peak shifts, disappearance, or formation of new peaks were observed in the FTIR spectrum of the mixture, suggesting the absence of strong chemical interactions or degradation. The findings indicate that Compritol ATO 888 and Favipiravir remain compatible, with only physical interactions contributing to the formulation. This confirms that Compritol ATO 888 can effectively function as a release-retarding agent in the formulation without compromising the chemical stability of Favipiravir, ensuring controlled drug release while maintaining the physicochemical properties of the active ingredient.

The FTIR spectrum of the FR and Crospovidone (crosslinked polyvinylpyrrolidone) mixture was analyzed to assess potential molecular interactions and confirm the compatibility of the super disintegrant with the active pharmaceutical ingredient (Figure 12). The characteristic peaks of Favipiravir remained unchanged, with a strong C=O stretching vibration observed at 1715 cm^−1^, confirming the stability of its keto-amide functional group. The amide N-H bending peak at 1625 cm^−1^ and the aromatic C=C stretching vibrations around 1550 cm^−1^ further validated the structural integrity of FR. Additionally, a broad O-H stretching band in the 3100–3500 cm^−1^ range was present, suggesting the potential for hydrogen bonding. Crospovidone exhibited its characteristic C=O stretching peak at 1650 cm^−1^, attributed to the pyrrolidone ring, along with C-N stretching vibrations in the 1200–1300 cm^−1^ range. Importantly, no significant peak shifts, disappearance, or formation of new peaks were observed in the FTIR spectrum of the FR-Crospovidone mixture, indicating the absence of strong chemical interactions. The results suggest that the interaction between FR and Crospovidone is primarily physical rather than chemical, ensuring the stability of the active ingredient. This confirms that Crospovidone can be safely incorporated as a superdisintegrant in the formulation without affecting the physicochemical integrity of Favipiravir, thereby facilitating rapid disintegration and improved drug dissolution in the final dosage form.

The FTIR spectrum of the Roflumilast (RT) and Crospovidone (crosslinked polyvinylpyrrolidone) mixture was analyzed to evaluate possible molecular interactions and ensure the compatibility of the superdisintegrant with the active pharmaceutical ingredient (Figure 13). The characteristic peaks of Roflumilast remained intact, with the strong C=O stretching vibration of the cyclic carboxylester group observed at 1730 cm^−1^, confirming the structural stability of the ester functionality. Additionally, the amide C=O stretching peak at 1665 cm^−1^ remained unaltered, while the aromatic C=C stretching vibrations at 1600 cm^−1^ validated the integrity of the benzamide core. The C-N stretching vibrations of the pyridine ring were also retained in the 1200–1300 cm^−1^ range, further confirming the absence of degradation or chemical interaction. Crospovidone exhibited its characteristic C=O stretching peak at 1650 cm^−1^, attributed to the pyrrolidone ring, along with C-N stretching vibrations in the fingerprint region. Importantly, the FTIR spectrum of the RT-Crospovidone mixture did not show any significant shifts, disappearance, or formation of new peaks, indicating that no strong chemical interactions or incompatibilities occurred. This suggests that the interaction between RT and Crospovidone is primarily physical rather than chemical, ensuring the stability of Roflumilast in the formulation. These findings confirm that Crospovidone can be safely incorporated as a superdisintegrant in the immediate-release layer without affecting the physicochemical integrity of Roflumilast, thereby facilitating rapid tablet disintegration and enhancing drug dissolution performance. Amorphous solid dispersion of excipients–drug and excipients to the drug was prepared by employing the solvent evaporation method. FR and RT solutions were prepared by dissolving 20 g of the sample, constituting 60% *w*/*w* drugs and 40% *w*/*w* excipients in 100 mL of methanol. The solvent was then evaporated using a rotary evaporator at 45 Celsius. The samples were then treated to remove any residual solvent.

To compare the stability of the amorphous state of RT and FR in each granule type, 10 g of each sample was kept in containers at 75% RH, with an ambient temperature of 25 °C, for the specified periods (2 weeks). The samples were then measured by FTIR upon removal of the lid at each time point.

The FTIR spectra of FR and RT after the preparation of amorphous solid dispersions (ASDs) using the solvent evaporation method were analyzed to assess any potential changes in their molecular interactions and structural integrity. The primary concern in ASD formation is whether hydrogen bonding, molecular dispersion, or interactions with excipients lead to significant peak shifts, disappearance, or the formation of new peaks (Figure 14). For FR, the characteristic carbonyl (C=O) stretching vibration at 1715 cm^−1^, associated with the keto-amide functional group, remained largely unchanged, indicating that no significant chemical transformation occurred (Figure 14). However, a slight broadening of the O-H stretching band (3100–3500 cm^−1^) was observed, which suggests the possible formation of hydrogen bonds with the excipients in the amorphous state. The amide N-H bending peak (1625 cm^−1^) and aromatic C=C stretching vibrations (1550 cm^−1^) were retained, confirming the molecular stability of FR.

For RT, the FTIR spectrum showed that the strong ester C=O stretching peak at 1730 cm^−1^ remained intact, confirming that the cyclic carboxylester functionality was not degraded (Figure 15). The amide C=O stretching peak at 1665 cm^−1^ and aromatic C=C peaks around 1600 cm^−1^ were also preserved. However, similar to FR, a broadening of the O-H stretching band in the 3100–3500 cm^−1^ range was observed, indicating potential hydrogen bonding interactions between RT and the excipients (Figure 16).

After two weeks of stability testing at 75% RH and 25 °C, the FTIR spectra of both FR and RT showed no significant new peak formations, no disappearance of characteristic peaks, and no major peak shifts beyond minor broadening in the hydrogen bonding region. This confirms that the amorphous solid dispersions maintained the structural integrity of both drugs, with only physical interactions occurring. The results indicate that the solvent evaporation method successfully transformed FR and RT into an amorphous state while maintaining their chemical stability and compatibility with the excipients.

## 3. Discussion

Fixed-dose combinations of drugs are beneficial for both patients and the health care system, but the current rationale of how formulations containing more than one drug work at the molecular level is still in its infancy. Mixing more than two drugs to formulate a novel pharmaceutical product is a complex procedure and formulating a successful multi-drug-containing pill continues to be a challenge for the researchers. Pharmaceuticals are often forced to delay the submission of multi-drug-containing pills because of formulation limitations and obstacles to the approvals imposed by regulators [14,15].

The dissolution and solution chemistry of the proposed multi-drug formulation is critical in making it purposeful and efficacious. The critical attributes of this combination were studied by evaluating the nature of dispersions with excipients, compatibilities, physical attributes, and dissolution properties.

Multi-drug formulations containing FR and RT can be promising combinations for treating COVID-19. FR treats SARS-CoV-2 by inhibiting viral replication, whereas RT suppresses a myriad of pro-inflammatory responses, thus limiting alveolar inflammation. The rationale for formulating such a combination was found to be cost-effective, industrially practicable, and compatible with entrapping the selected drugs (RT and FR) as the physicochemical parameters were found to be within the set criteria. The combination was proposed with an immediate-release layer and a controlled-release layer simultaneously. Different grades of functional excipients are employed for definite functions [16].

Stability assessment is crucial to ensure the long-term viability and robustness of the fixed-dose combination (FDC) of Roflumilast (RT) and Favipiravir (FR). To evaluate the stability of the formulation under varying environmental conditions, ICH Q1A(R2) stability guidelines were followed, conducting both accelerated (40 °C ± 2 °C/75% RH ± 5% RH) and long-term (25 °C ± 2 °C/60% RH ± 5% RH) stability studies over six months. The primary objective was to assess the chemical and physical stability of the APIs and excipients, ensuring they maintained their integrity without degradation or adverse interactions. Analytical methods such as HPLC and FTIR were employed to monitor drug content, degradation products, and potential interactions between the APIs and excipients. No significant degradation of RT and FR was observed, with drug content remaining within the acceptable range of 90–110% of the initial value. The dissolution profile of both layers remained consistent throughout the stability period, confirming that neither immediate- nor controlled-release mechanisms were compromised. Moisture uptake studies indicated minimal hygroscopicity, which was mitigated by the selection of excipients like Compritol ATO888, which provided a protective lipid matrix, and Klucel HXF, which maintained gel integrity even under high humidity. Physical attributes, including tablet hardness, friability, weight variation, and disintegration time, remained within pharmacopeial limits, with no evidence of tablet softening, layer separation, or changes in organoleptic properties. These findings confirm that the FDC formulation is chemically and physically stable under both normal and accelerated conditions, supporting its potential shelf life of at least 24 months. The excipients used in the formulation, including Klucel HXF, Klucel EXF, Compritol 888 ATO, microcrystalline cellulose, calcium carbonate, PVP, sodium stearyl fumarate, colloidal silica oxide, magnesium stearate, and colorants, were analyzed individually to ensure their spectral compatibility with the APIs.

The release of API from both layers is regulated using different grades of Klucel (Hydroxypropyl Cellulose). Hydroxypropyl cellulose with fine particle size (Klucel HXF) is known for its swelling behaviors and its applications in controlled-release matrices (Figure 17). By incorporating such polymer, drugs tend to stay within a core and release by opting for a phenomenon of diffusion and erosion. It imparts good compaction properties to powders and is thermodynamically stable, whereas low-molecular-weight Klucel EXF are employed for their good compactions and water carrying properties for early release of drugs from tablets.

The range and efficiency of erosion are different for different grades of Klucel. Lower-molecular-weight grades tend to erode at a faster rate than the higher ones [17].

Compritol^®^ 888 ATO is a hydrophobic excipient that forms an inert matrix in controlled-release formulations, allowing drug diffusion without swelling or erosion. Drug release follows first-order kinetics. The tablets also contain bulking agents, disintegrants, flow enhancers, and lubricants. F1 and F2 were optimized with different levels of Klucel HXF and Compritol^®^ 888 ATO in Layer 2 to regulate FR release, while Layer 1 includes Klucel EXF and Crospolyvinylpyrrolidone to ensure the immediate release of RT [17].

Drug–excipient compatibility studies using FTIR and physical testing confirmed no significant interactions between RT, FR, and excipients. Both formulations (F1 and F2) were inert to incompatibilities and met pharmacopeial limits for hardness, friability, and weight variation, with F1 showing better compaction properties. A validated HPLC method was developed to analyze assay and dissolution behaviors, demonstrating linearity, accuracy, and precision. Calibration curves for RT and FR showed linear relationships across a 50–150% concentration range. The recoveries were within 90–110%, and the %RSD values were below 2%, confirming the method’s accuracy and reproducibility. Intra-day analysis further validated the precision of the developed method [17]. The robustness and ruggedness of the proposed method were demonstrated through minimal variations in experimental conditions, with no significant changes observed in peak areas, recoveries, or retention times for RT and FR. Assay and content uniformity were maintained, with F1 showing better uniformity than F2. Dissolution studies revealed the immediate release of RT and controlled release of FR over 12 h, with F1 proving more efficient due to its higher Klucel EXF content. F2 exhibited dose dumping of FR due to excess Compritol^®^ 888 ATO, making its dissolution behavior non-uniform. The solvent evaporation method was used to prepare amorphous solid dispersions, with no FTIR shifts detected. The F2 formulation shows promise for a fixed-dose combination of RT and FR for COVID-19 treatment, with both APIs demonstrating favorable release kinetics.

## 4. Materials and Methods

### 4.1. Materials

#### 4.1.1. Preparation for Fixed-Dose Combination

FR was obtained as a gift sample from Hubei Biocause Pharmaceutical Co., Ltd., Jingmen, China. Roflumelast was obtained as a donation from Vision Pharmaceuticals Islamabad, Pakistan. Klucel HXF was obtained as a sample from Ashland Corporation Malaysia, Johor Bahru, Malaysia. Compritol-888 ATO was obtained as a donation from Gattefossé, Paramus, NJ, USA. The gift samples of PVP, microcrystalline cellulose, and calcium carbonate were provided by Vision Pharmaceuticals, Islamabad, Pakistan. Sodium stearyl fumarate, colloidal silica oxide, magnesium stearate, and royal blue color were obtained as a donation from Biolabs Pharmaceuticals, Islamabad, Pakistan. Further equipment was used for the preparation of FDC, including a weigh balance, sieves, a double cone mixer, and a bilayer rotary double press.

#### 4.1.2. Evaluation of Release Kinetics

The chemicals and reagents used in the evaluation of this FDC are acetonitrile, sodium perchlorate, triethylamine, methanol, and distilled water. Further equipment used for the evaluation of FDC includes a weighing balance, sonicator, Shimadzu FTIR, Shimadzu HPLC, and Fibrillator.

### 4.2. Dosage Form Development and Physicochemical Evaluation

#### 4.2.1. Preparation of Fixed-Dose Combination

All the preparation of fixed dose combinations are stated in Table 12.

**Table 12 pharmaceuticals-18-00590-t012:** Formulations for fixed-dose combination (FDC) of FR and Roflumelast. This table presents the composition of two formulations (F1 and F2) designed for a fixed-dose combination (FDC) of FR and Roflumelast, highlighting the function and quantity of each component in the immediate-release (1st layer) and controlled-release (2nd layer) layers. FR, an antiviral agent, is incorporated into the controlled-release layer, while Roflumelast, a phosphodiesterase-4 inhibitor, is included in the immediate-release layer (Figure 18). Klucel HXF and Klucel EXF serve as sustained-release and immediate-release polymers, respectively. Other excipients include Compritol 888 Ato as a release-retarding agent, microcrystalline cellulose and calcium carbonate as bulking agents, PVP as a binder, colloidal silica oxide as a flow enhancer, and sodium stearyl fumarate and magnesium stearate as lubricants. The colorant used in the formulation is royal blue, added to the controlled-release layer.

		Formulation 1 (F1)		Formulation 2 (F2)	
Components	Functions	1st Layer (Immediate) mg	2nd Layer (Controlled) mg	1st Layer (Immediate) mg	2nd Layer (Controlled) mg
FR	Anti-viral	-	1600	-	1600
RT	Phosphodiesterase-4 inhibitors	0.5	-	0.5	-
Klucel HXF	Sustained-release polymer	-	200	-	150
Klucel EXF	Immediate-release polymer	50	-	30	-
Compritol 888 Ato	Release-retarding agent	-	5	-	10
Microcrystalline Cellulose	Bulking agent	141	-	161	-
Calcium Carbonate	Bulking agent	-	170.5	-	205.5
Sodium Stearyl Fumarate	Lubricant	3	-	3	-
PVP	Binder	5	10	5	20
Colloidal Silica Oxide	Flow enhancer	0.5	2.5	0.5	2.5
Magnesium Stearate	Lubricant	-	10	-	10
Royal blue color	Colorant	-	2	-	2
Total		200	2000	200	2000

**Figure 18 pharmaceuticals-18-00590-f018:**
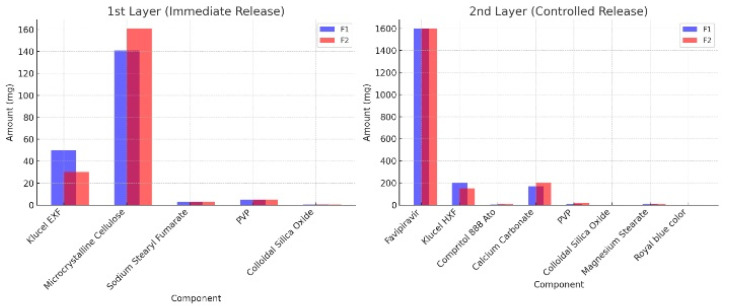
Schematic representation of formulation F1 (immediate release) and F2 (controlled release) layers. This figure illustrates the structural composition of the two-layered formulations F1 and F2, showing the distribution of APIs and excipients in the immediate-release and controlled-release layers. A graphical representation comparing the quantitative differences in excipients used in formulation 1 (F1) and formulation 2 (F2), highlighting variations in polymer concentrations, bulking agents, and release-modifying agents. A diagram explaining the dissolution and release behavior of FR and Roflumelast from the respective layers, demonstrating the effect of polymer concentration and release-retarding agents on drug kinetics.

#### 4.2.2. Blending Procedure

Layer 1

Klucel EXF and microcrystalline cellulose are sieved with mesh # 20 and mixed (Figure 19(right)).Further Crospolyvinylpyrrolidone is added and mixed in the main bulk.Colloidal silicon dioxide is sieved, added to the main bulk, and mixed.Roflumelast is added geometrically to the main bulk and mixed.The main bulk is then lubricated by the addition and mixing of sodium stearyl fumarate.

Layer 2

FR and Compritol 888 Ato are sieved with mesh # 20 and mixed to form a release-retarding layer around FR (Figure 19(left)).Klucel HXF is sieved with mesh # 20 and is added to the main bulk.Calcium carbonate, royal blue color, and colloidal silicon dioxide are sieved with mesh # 20 and mixed with the main bulk.Crosspolyvinylpyrrolidone is further added to the main bulk.The main bulk is then lubricated by addition and mixing of magnesium stearate [18].

#### 4.2.3. Tableting

Rotary press ZP-25 for bilayer tableting was used for molding both layers into a unified single tablet with two distinctive layers.

The optimization of the immediate-release layer of RT was carefully designed to achieve rapid and complete dissolution, ensuring optimal bioavailability. Klucel EXF (low viscosity hydroxypropylcellulose) was selected due to its superior binding properties and its ability to enhance drug dispersion in aqueous media. Various concentrations (1% to 5% *w*/*w*) were evaluated, with the optimal concentration balancing tablet integrity and rapid disintegration. The selection process also involved comparing different disintegrants, such as sodium starch glycolate and croscarmellose sodium, where Klucel EXF demonstrated the most efficient immediate drug release without compromising mechanical strength. Dissolution studies conducted in USP Type II apparatus using pH 1.2 and pH 6.8 media confirmed that Klucel EXF facilitated rapid hydration and disintegration, allowing RT to dissolve completely within 30 min. Unlike other binders, Klucel EXF does not form a gel matrix, ensuring that the drug release mechanism remains truly immediate. Additionally, stability testing confirmed that Klucel EXF did not interact with RT, preserving drug integrity and ensuring uniform content distribution. The overall formulation strategy, incorporating Klucel EXF, was critical in optimizing dissolution kinetics while maintaining the stability and uniformity of the IR layer.

#### 4.2.4. Evaluation of Release Kinetics

##### High-Performance Liquid Chromatography

The concentration of drugs was determined using HPLC. The system was a Schimadzu high-performance liquid chromatography (HPLC) system with a UV detector and a C18 column (4.6 mm 25 cm) with a particle size of 5 m. The injection volume was 20 mL, and absorbance was measured at a wavelength of 275 nm. Analyses were carried out at a column temperature of 30 degrees Celsius for a total run time of 10 min at a flow rate of 1 mL per minute. Acetonitrile and buffer (sodium perchlorate 2 g + triethylamine 5 mL) were employed in a 60:40 (*v*/*v*) ratio as the mobile phase, while water was utilized as the stationary phase [19].

##### Assay of Formulation

Preparation of stock solution A:

Using a volumetric flask with a 50 mL capacity, accurately weigh about 50 mg of RT working standard. Add approximately 30 mL of mobile phase and sonicate to dissolve and then add additional mobile phase to obtain the volume up to the required level.

Take 1 mL of the filtrate and place it in a 100 mL volumetric flask, filling the flask to the specified capacity with the mobile phase.

Preparation of stock solution B:

Place about 1600 mg of FR working standard in a 50 mL volumetric flask and weigh it accurately. Add approximately 30 mL of mobile phase and sonicate to dissolve and then add additional mobile phase to obtain the volume up to the required level.

Preparation of standard solution:

Take exactly 0.5 mL of stock solution A and 0.5 mL of stock solution B and place them in a 50 mL volumetric flask. Fill the flask to the mark with the mobile phase to produce a final concentration of 0.32 mg/mL for FR and 0.0001 mg/mL for RT.

Preparation of sample solution:

Weigh 20 tablets to obtain the average weight per tablet. Grind them finely, then weigh accurately sample powdered tablets equivalent to average weight and transfer them into a 100 mL volumetric flask. Add about 30 mL of diluent and sonicate for 10 min, and then make the volume up to the mark with diluent and mix.

Then, take 1 mL of the filtrate into another 50 mL volumetric flask so that the final concentration becomes 0.32 mg/mL of FR and 0.0001 mg/mL of RT. Filter the solution through a 0.45 µm syringe filter.

Procedure:

Separately inject equal volumes (about 20 µL) of the standard preparation and the assay preparation, record the chromatograms, and measure the responses for the major peaks.

##### Dissolution of Formulations

Dissolution under non-sink parameters was performed on apparatus II. One tablet was added to 500 mL of 0.1 N HCl contained in individual bowls of dissolution apparatus. The dissolved concentrations were determined at 30 min, 4 h, 8 h, and 12 h.

Preparation of stock solution A:

Accurately weigh about 50 mg of RT working standard in a 50 mL volumetric flask. Add about 30 mL of mobile phase and sonicate to dissolve, and then make the volume up to the mark with the mobile phase. Take 1 mL of filtration into another 100 mL volumetric flask and make the volume up to the mark with the mobile phase.

Preparation of stock solution B:

Accurately weigh about 1600 mg of FR working standard in a 50 mL volumetric flask. Add about 30 mL of mobile phase and sonicate to dissolve, and then make the volume up to the mark with the mobile phase.

Preparation of standard solution:

Accurately take 0.5 mL of stock solution A and 0.5 mL of stock solution B in a 50 mL volumetric flask and then make the volume up to the mark with mobile phase to obtain a final concentration of 0.32 mg/mL for FR and 0.0001 mg/mL for RT.

Sample Solution:

Dissolution under non-sink parameters was performed on apparatus II. One tablet was added to 500 mL of 0.1 N HCl contained in individual bowls of dissolution apparatus. The samples were withdrawn at 30 min, 4 h, 8 h, and 12 h to quantify concentrations of controlled-release FR and immediate-release RT through two distinctive layers of tablets.

Procedure:

Now, inject both sample solutions and standard solutions into the chromatograph after setting the chromatographic conditions, mobile phase same as mentioned in the assay, and record the retention time and area occupied by the major peak in both chromatograms.

##### Physical Attributes

Weight Variation

Ten tablets were taken randomly from both formulations F1 and F2 and were weighed individually to check for weight variation. Weight variation specification was considered as per USP [20].

Content Uniformity

Weigh precisely 10 tablets one at a time and determine the amount of drug ingredient included in each tablet’s contents, expressed as a percentage of the label claim’s age. Identify and calculate the acceptance value (AV). It is necessary to maintain dose consistency by lowering the acceptance value of the first 10 dosage units to lower than or equal to L1. This need must be satisfied for the requirement to be considered met. If the acceptance value is greater than L1, test the next 20 dose units and compute the acceptance value for each one. If the acceptance value is lower than L1, the process should be repeated. If the ultimate acceptance value of all 30 dosage units is lower than or equal to L1, and if no individual content of a dosage unit is less than (1 − L2 0.01) M or higher than (1 + L2 0.01) M when the acceptance value is computed under either content uniformity or mass variation, it is sufficient considering nothing else, and the values of L1 and L2 are 15.0 and 25, respectively [21].

Hardness and Friability

For tablets with a unit mass equal to or less than 650 mg, acquire a sample of complete tablets weighing 6.5 g or less to determine the unit mass. For tablets with a unit mass greater than 650 mg, gather a sample of ten whole tablets to be weighed and tested. Before testing, it is necessary to thoroughly dedust the tablets. Weigh the tablet sample accurately, and then place the tablets in the drum to be weighed. Extraction of the pills is accomplished by rotating the drum 100 times. Remove any loose dust from the tablets, as carried out earlier, and weigh them appropriately.

Drug Carrier–Excipients Compatibility Testing

The drug and excipient compatibility studies were carried out by checking the physical description and using the Fourier transform infrared spectroscopy analytical technique. The interaction studies were carried out to ascertain any incompatibilities of the drug with the excipients used in the preparation of multi-drug-containing tablets of FR and RT solid dispersions. The disappearance of an absorption peak, the appearance of new peaks, or a reduction purity index are markers of the existence of incompatibilities between the API and the excipient under study.

An FT-IR spectrophotometer was used for infrared investigations of samples. About 4–5 mg of the specimen was mixed with dry potassium bromide (KBr), and the specimen was examined at transmission mode over the wave number range of 4000–400 cm^−1^. FT-IR spectroscopy employing a potassium bromide-pressed disk was conducted on a Frontier spectrometer (Shimadzu, FTIR-000180, Kyoto, Japan). Approximately, 10 mg of drug excipient mixture specimen, 10 mg of drug, and 100 mg of KBr were weighed, sized in an agate mortar, and pressed for 10 min at approximately 10 tones/cm^2^ to form a semitransparent pellet which lets light be transmitted to the detector [22].

##### Preparation of Amorphous Solid Dispersions

Amorphous solid dispersion of excipients–drug and excipients–2 (drug) was prepared by employing the solvent evaporation method. FR and RT solutions were prepared by dissolving 2 g of sample constituting 60% *w*/*w* excipients and 40% *w*/*w* of drugs in 10 mL of methanol. The solvent was then evaporated using a rotary evaporator at 45 °C. The samples were then treated to remove any residual solvent.

The stability of RT and FR in their amorphous states in each granule type was evaluated by maintaining two grams of each sample in containers at 75% relative humidity and an ambient temperature of 25 °C for the periods indicated (2 weeks at a maximum). The samples were measured by FTIR as soon as the lid was removed at each time point, and the results were recorded immediately [2].

## 5. Conclusions

A novel fixed-dose combination (FDC) of roflumilast (RT) and favipiravir (FR) was successfully developed with satisfactory physicochemical properties, compatibility, and stability. The optimized formulation demonstrated a well-regulated immediate and controlled-drug-release profile, making it a promising candidate for improving treatment adherence and therapeutic outcomes in COVID-19 patients. By reducing the medication burden, this FDC product could enhance patient compliance and offer a more pharmaco-economical alternative to conventional individual tablets.

Moreover, the FDC developed has the potential to lower production and administrative costs, facilitating broader accessibility to treatment in resource-limited settings such as Pakistan. The newly established analytical method for the simultaneous quantification of RT and FR was found to be simple, precise, and economical, enabling efficient quality control in pharmaceutical manufacturing and routine laboratory analysis.

The conclusion emphasizes that the fixed-dose combination (FDC) of Roflumilast (RT) and Favipiravir (FR) has the potential to reduce medication load and improve patient compliance, a critical factor in effective treatment outcomes. Several studies have demonstrated that simplified dosing regimens and reduced pill burden significantly enhance adherence, particularly in chronic and infectious diseases. (i) Combining immediate-release RT and controlled-release FR into a single tablet ensures optimized therapeutic levels with fewer doses per day. (ii) This eliminates the complexity of taking multiple pills at different times, which is a well-known factor in non-adherence. (iii) Studies in HIV, cardiovascular diseases, and diabetes have shown that FDCs improve adherence by offering a single-pill alternative to multiple medications. (iv) This is particularly beneficial for elderly patients or those with comorbidities, who are often prescribed multiple drugs. (v) Patients are more likely to follow prescribed regimens when they are easy to manage. (vi) Ensuring consistent drug levels, the FDC may lead to better clinical outcomes and reduced risk of disease progression.

Future directions include (but are not limited to) the following:A.In vivo pharmacokinetic and pharmacodynamic studies to assess the bioavailability and therapeutic efficacy of the FDC formulation in comparison to individual drug administration.B.Long-term stability studies ensure the robustness of the formulation under varying environmental conditions.C.Clinical trials to evaluate the safety, efficacy, and patient adherence associated with this combination therapy in COVID-19 and related viral infections.D.Exploration of alternative drug delivery systems, such as nanocarriers or orodispersible formulations, to further optimize the pharmacological performance and patient convenience of RT-FR combinations.E.Regulatory pathway assessment to facilitate the approval process and eventual market implementation of this novel FDC.

These future directions will help establish a solid scientific foundation for the widespread adoption of this formulation, contributing to more effective and accessible antiviral therapies.

## Figures and Tables

**Figure 1 pharmaceuticals-18-00590-f001:**
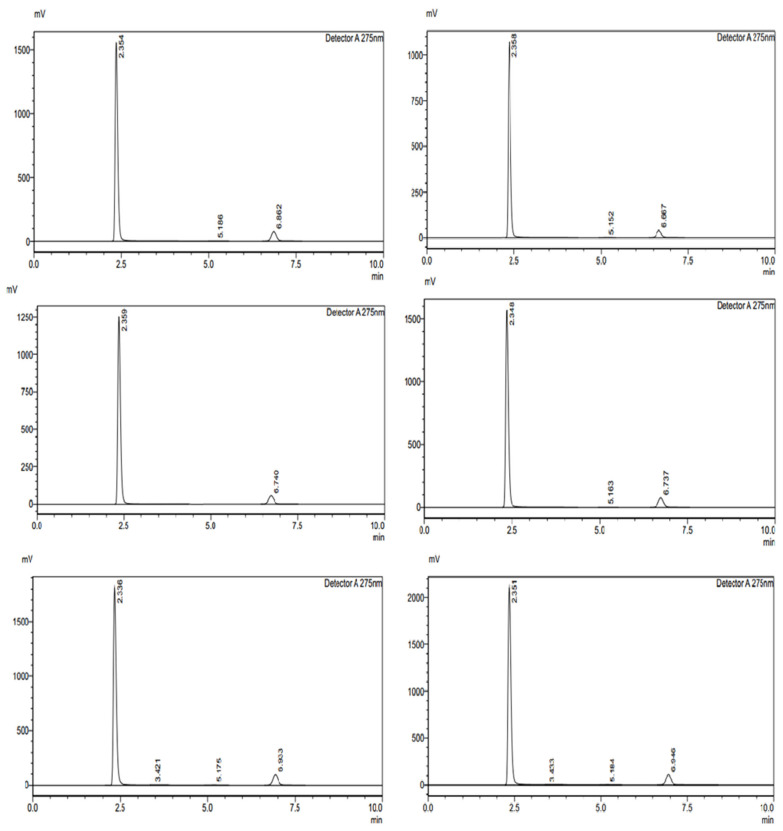
Linearity standard, linearity 50%, linearity 75%, linearity 100%, linearity 125%, and 150%. This figure represents the linearity of FR by plotting the standard and sample responses against concentration levels, illustrating the method’s ability to provide a proportional response across the tested range.

**Figure 2 pharmaceuticals-18-00590-f002:**
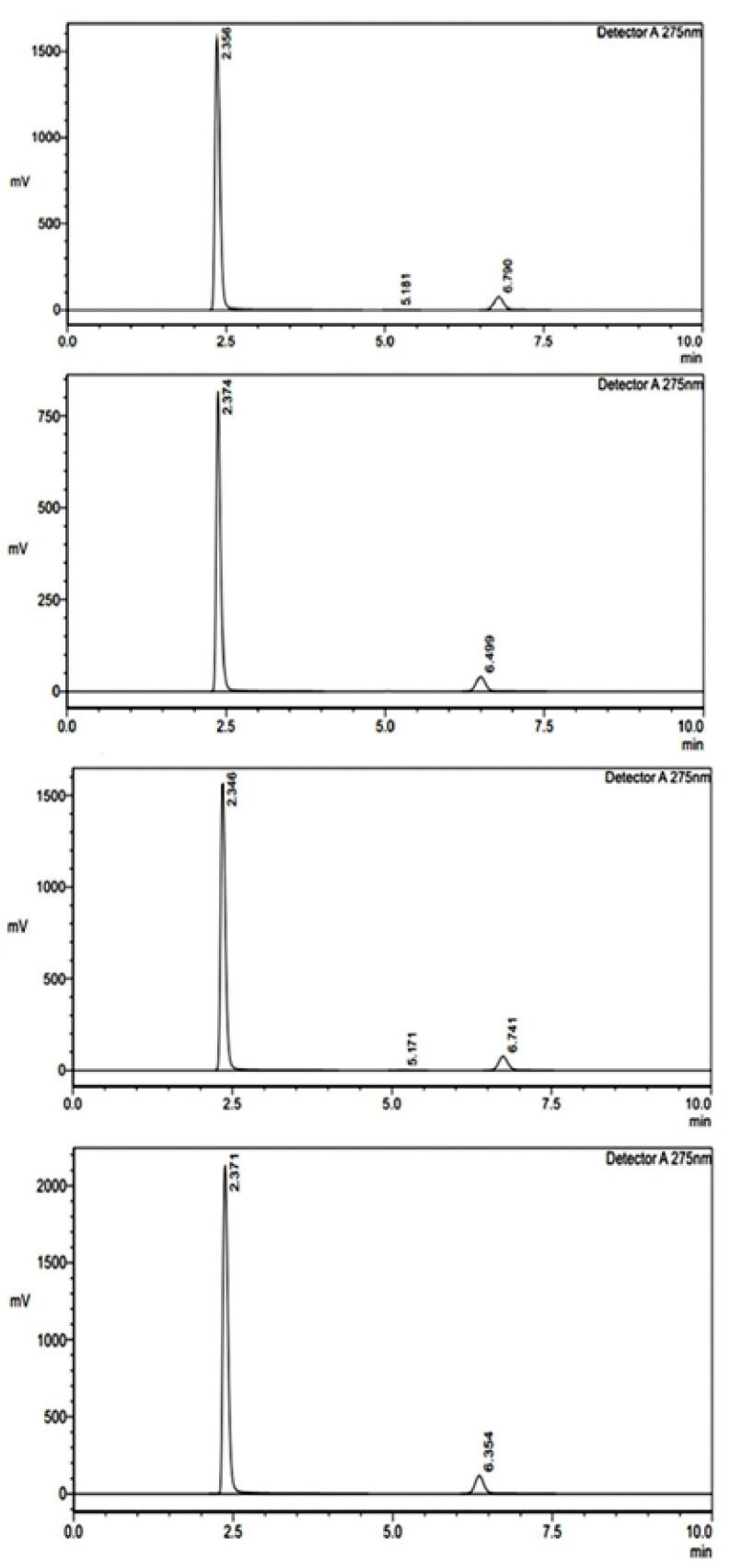
Accuracy and recovery standard, 50%, 100% and 150%.

**Figure 3 pharmaceuticals-18-00590-f003:**
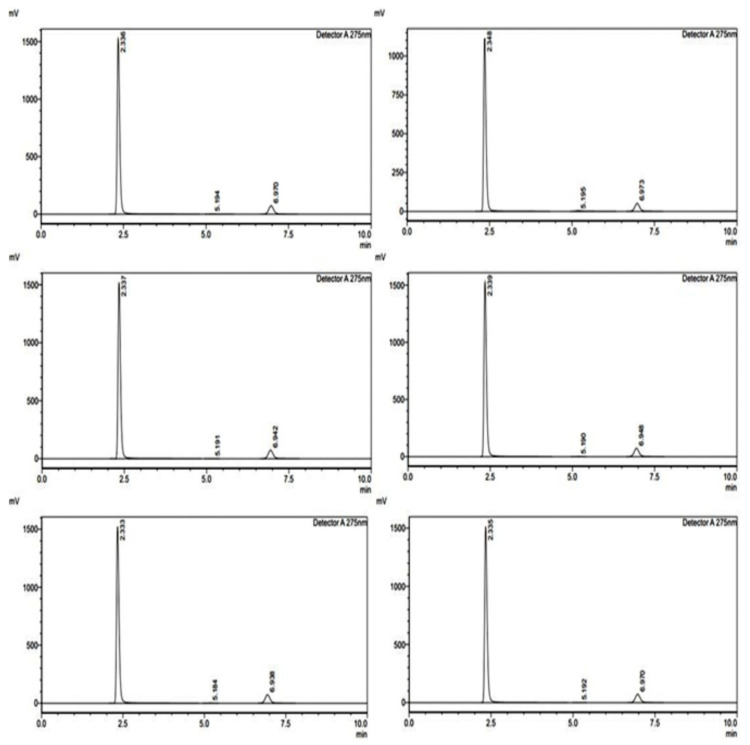
Method precision standard Day 1, Method precision sample Day 1, Method precision standard Day 2, Method precision sample Day 2, Method precision standard Day 3, Method precision sample Day 3.

**Figure 4 pharmaceuticals-18-00590-f004:**
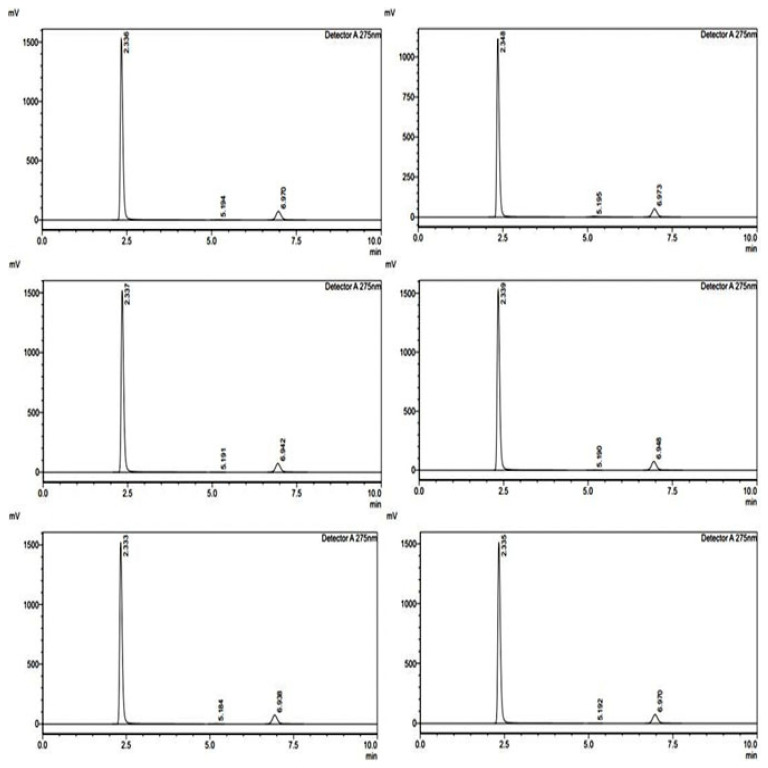
Ruggedness standard Analyst 1, ruggedness sample Analyst 1, ruggedness standard Analyst 2, ruggedness sample Analyst 2, ruggedness standard Analyst 3, ruggedness sample Analyst 3.

**Figure 5 pharmaceuticals-18-00590-f005:**
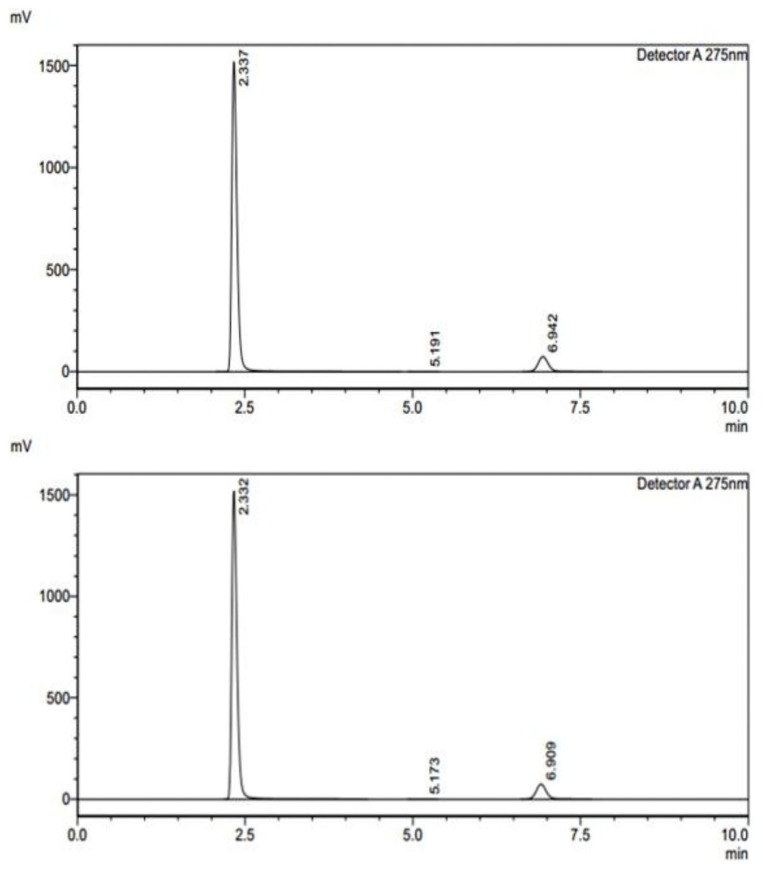
Standard for assay analysis of F1, sample for assay analysis of F1.

**Figure 6 pharmaceuticals-18-00590-f006:**
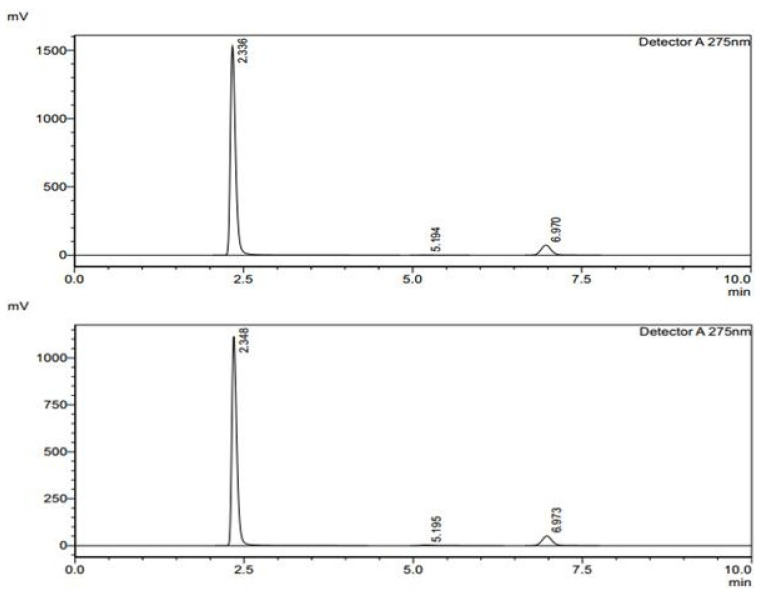
Standard for dissolution of F1, sample for dissolution of F1.

**Figure 7 pharmaceuticals-18-00590-f007:**
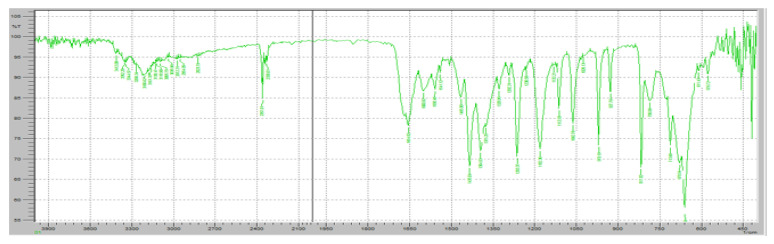
FTIR scan for FR.

**Figure 8 pharmaceuticals-18-00590-f008:**
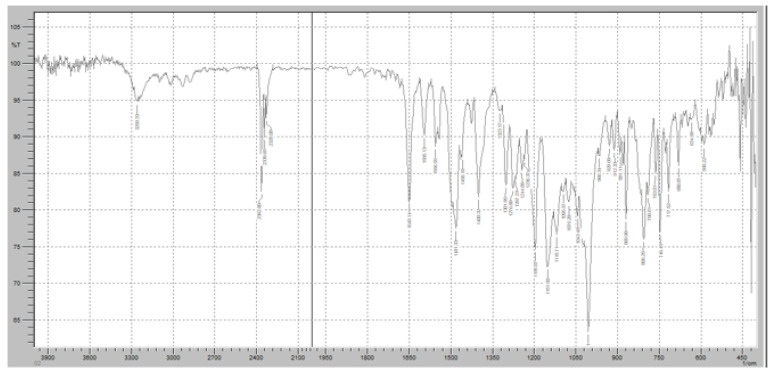
FTIR scan for Roflumelast.

**Figure 9 pharmaceuticals-18-00590-f009:**
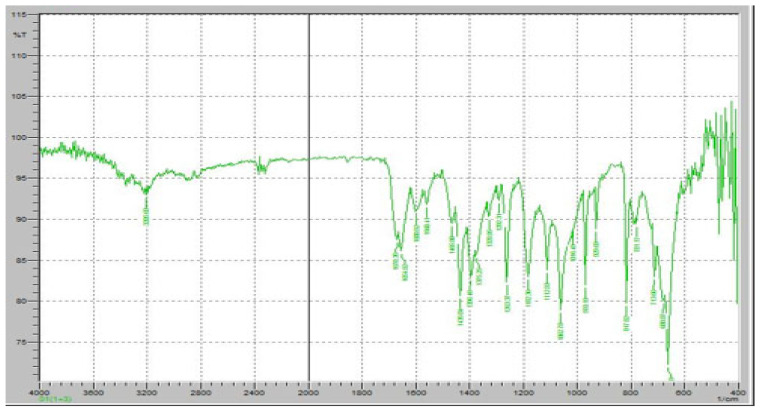
FTIR scan for FR and Klucel HXF mixture.

**Figure 10 pharmaceuticals-18-00590-f010:**
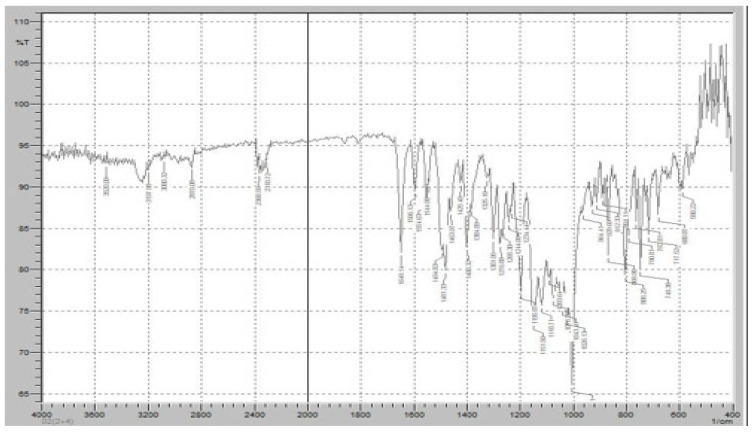
FTIR scan for Roflumelast and Klucel EXF mixture.

**Figure 11 pharmaceuticals-18-00590-f011:**
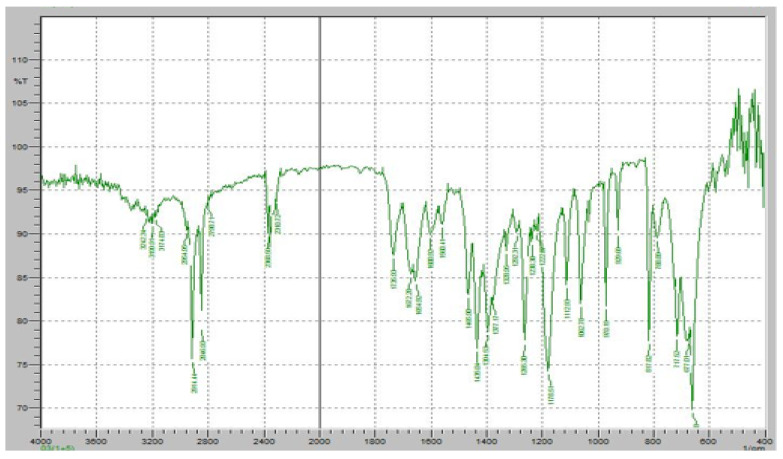
FTIR scan for FR and Compritol Ato888 mixture.

**Figure 12 pharmaceuticals-18-00590-f012:**
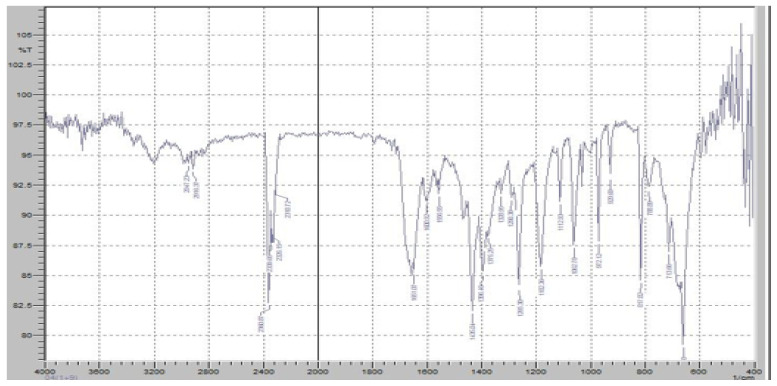
FTIR scan for FR and Crospolyvinylpyrrolidone mixture.

**Figure 13 pharmaceuticals-18-00590-f013:**
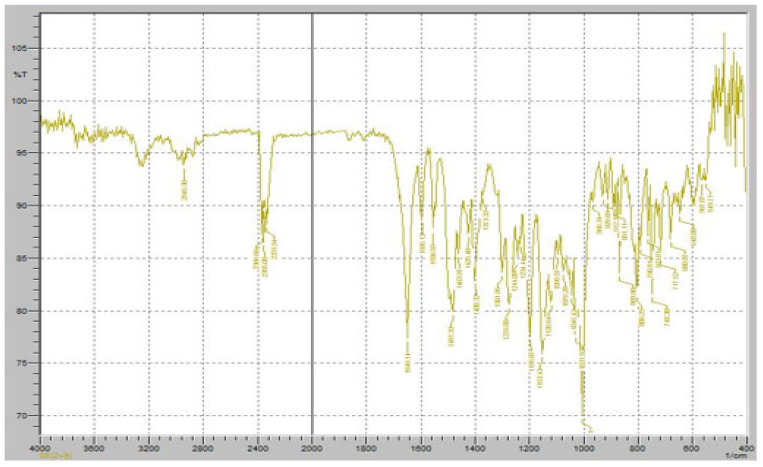
FTIR scan for Roflumelast and Crospolyvinylpyrrolidone mixture.

**Figure 14 pharmaceuticals-18-00590-f014:**
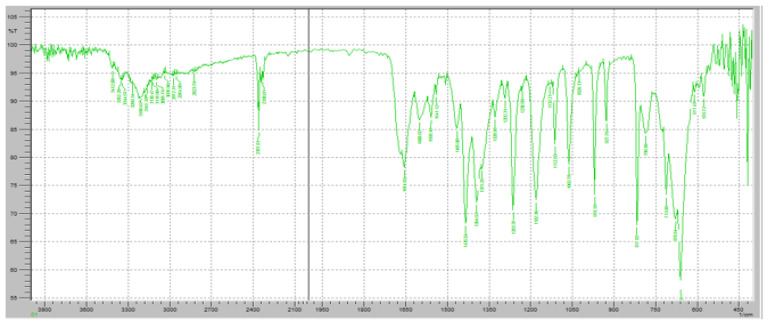
FTIR scan for FR.

**Figure 15 pharmaceuticals-18-00590-f015:**
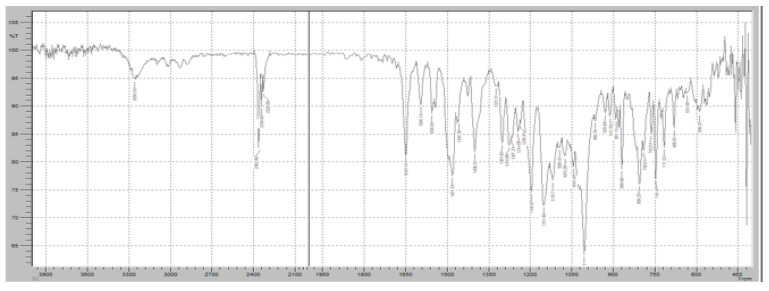
FTIR scan for Roflumelast.

**Figure 16 pharmaceuticals-18-00590-f016:**
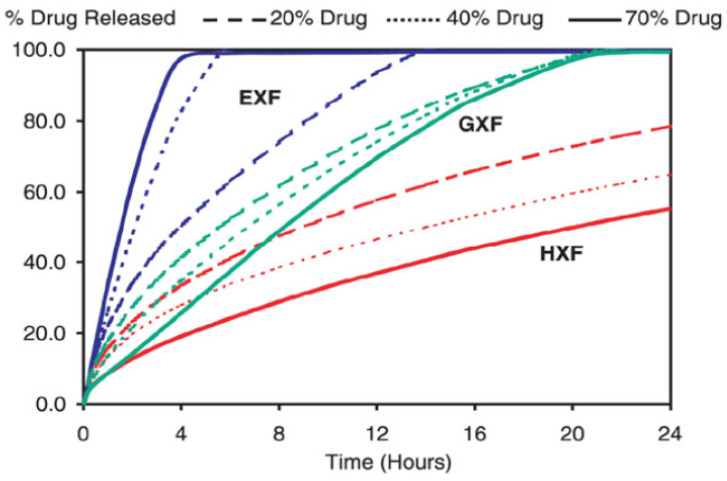
Klucel’s molecular weight affects drug release concerning time.

**Figure 17 pharmaceuticals-18-00590-f017:**
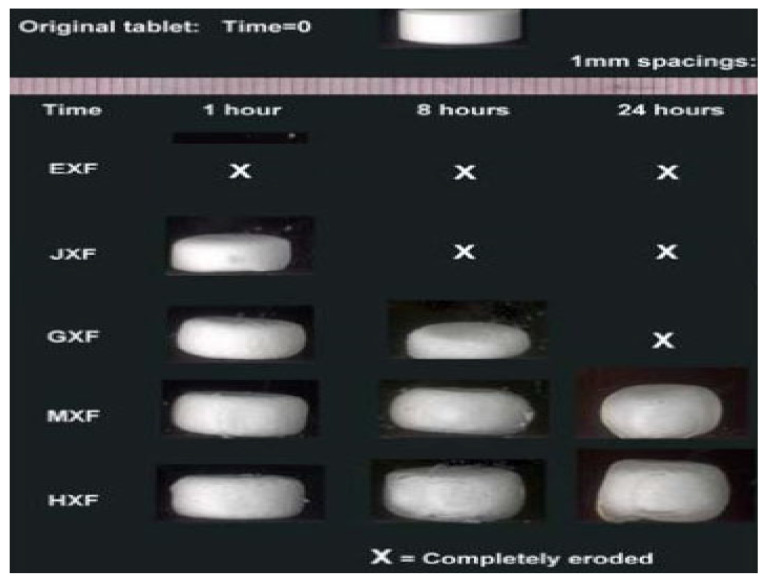
Swelling of tablets in dissolution media.

**Figure 19 pharmaceuticals-18-00590-f019:**
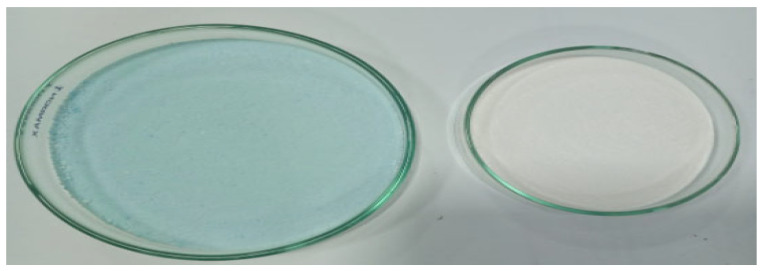
Layer 2 blue-colored and layer 1 white-colored.

**Table 1 pharmaceuticals-18-00590-t001:** Linearity for FR. This table presents the linearity evaluation for FR across different concentration levels (50%, 75%, 100%, 125%, and 150%). The standard and sample responses are recorded to determine the correlation between concentration and response. The percentage response reflects the accuracy of the analytical method, demonstrating the expected proportional increase in response with increasing concentration.

Linearity	Standard Response	Sample Response	Percentage Response
50%	8,233,034.67	4,127,632	50.14
75%		6,456,298	78.42
100%		8,216,768	99.80
125%		10,051,211	122.08
150%		12,201,096	148.20

**Table 2 pharmaceuticals-18-00590-t002:** Linearity for RT. This table presents the linearity evaluation for RT at different concentration levels (50%, 75%, 100%, 125%, and 150%). The standard and sample responses are recorded to assess the correlation between concentration and response. The percentage response indicates the method’s accuracy and reliability in maintaining a linear relationship between analyte concentration and instrument response.

Linearity	Standard Response	Sample Response	Percentage Response
50%	769,962.00	387,597	50.34
75%		563,203	73.15
100%		769,345	99.92
125%		982,867	127.65
150%		1,164,820	151.28

**Table 3 pharmaceuticals-18-00590-t003:** Accuracy and recovery data. This table presents the accuracy and recovery data for FR and RT at three concentration levels: 50%, 100%, and 150%. The standard mean response, measured sample response, and percentage recovery were recorded to assess the accuracy of the analytical method. The recovery percentage for FR ranged from 49.88% to 50.00% at 50% concentration, 101.89% to 102.16% at 100%, and 149.48% to 149.95% at 150%, demonstrating a high level of precision. Similarly, RT showed recoveries ranging from 54.31% to 54.59% at 50%, 102.15% to 102.35% at 100%, and 148.42% to 148.84% at 150%, indicating consistency in quantification. These results confirm the accuracy of the analytical method in determining the drug content at varying concentrations.

Sample	Standard (Mean)	Measured (Sample)	Concentration (%)	Analyte	Recovery (%)
50%	8,061,208.0	4,027,053.0	50%	FR	49.96
50%	8,061,208.0	4,028,347.0	50%	FR	49.97
50%	8,061,208.0	4,020,581.0	50%	FR	49.88
50%	8,061,208.0	4,021,683.0	50%	FR	49.89
50%	8,061,208.0	4,025,900.0	50%	FR	49.94
50%	8,061,208.0	4,030,829.0	50%	FR	50.0
100%	8,061,208.0	8,218,083.0	100%	FR	101.95
100%	8,061,208.0	8,220,843.0	100%	FR	101.98
100%	8,061,208.0	8,213,405.0	100%	FR	101.89
100%	8,061,208.0	8,229,584.0	100%	FR	102.09
100%	8,061,208.0	8,235,672.0	100%	FR	102.16
100%	8,061,208.0	8,223,300.0	100%	FR	102.01
150%	8,061,208.0	12,049,505.0	150%	FR	149.48
150%	8,061,208.0	12,056,936.0	150%	FR	149.57
150%	8,061,208.0	12,080,875.0	150%	FR	149.86
150%	8,061,208.0	12,087,897.0	150%	FR	149.95
150%	8,061,208.0	12,086,233.0	150%	FR	149.93
150%	8,061,208.0	12,082,560.0	150%	FR	149.89
50%	753,383.67	409,904.0	50%	RT	54.41
50%	753,383.67	409,507.0	50%	RT	54.36
50%	753,383.67	409,179.0	50%	RT	54.31
50%	753,383.67	410,058.0	50%	RT	54.43
50%	753,383.67	410,776.0	50%	RT	54.52
50%	753,383.67	411,301.0	50%	RT	54.59
100%	753,383.67	770,408.0	100%	RT	102.26
100%	753,383.67	769,841.0	100%	RT	102.18
100%	753,383.67	769,558.0	100%	RT	102.15
100%	753,383.67	770,531.0	100%	RT	102.28
100%	753,383.67	771,078.0	100%	RT	102.35
100%	753,383.67	770,095.0	100%	RT	102.22
150%	753,383.67	1,118,141.0	150%	RT	148.42
150%	753,383.67	1,119,127.0	150%	RT	148.55
150%	753,383.67	1,120,421.0	150%	RT	148.72
150%	753,383.67	1,121,369.0	150%	RT	148.84
150%	753,383.67	1,119,673.0	150%	RT	148.62
150%	753,383.67	1,119,708.0	150%	RT	148.62

**Table 4 pharmaceuticals-18-00590-t004:** Method precision data. This table presents the method precision data for FR and RT across three consecutive days. The standard and sample areas were measured, and the percentage recovery was calculated to evaluate intra-day and inter-day precision. The results for FR showed percentage values ranging from 99.77% to 100.68%, while RT exhibited values between 99.92% and 100.71%, confirming the high reproducibility and reliability of the analytical method. The minimal variability across days demonstrates the method’s consistency in quantifying both active pharmaceutical ingredients (APIs).

Day	Standard/Sample	Area (Standard)	Area (Sample)	Analyte	Percentage (%)
Day 1	Std 1	8,196,278	8,179,511.0	FR	99.81
Day 1	Std 2	8,205,135	8,182,452.0	FR	99.85
Day 1	Std 3	8,183,856	8,182,830.0	FR	99.85
Day 2	Std 1	8,179,792	8,187,300.0	FR	100.0
Day 2	Std 2	8,197,768	8,242,848.0	FR	100.68
Day 2	Std 3	8,183,856	8,210,166.0	FR	100.28
Day 3	Std 1	8,157,856	8,163,301.0	FR	99.97
Day 3	Std 2	8,161,651	8,166,868.0	FR	100.01
Day 3	Std 3	8,178,636	8,147,267.0	FR	99.77
Day 1	Std 1	770,065	770,020.0	RT	99.93
Day 1	Std 2	771,646	775,382.0	RT	100.63
Day 1	Std 3	769,952	774,907.0	RT	100.56
Day 2	Std 1	768,780	770,020.0	RT	100.01
Day 2	Std 2	771,432	775,382.0	RT	100.71
Day 2	Std 3	769,575	774,907.0	RT	100.65
Day 3	Std 1	766,633	767,044.0	RT	100.0
Day 3	Std 2	767,739	767,444.0	RT	100.05
Day 3	Std 3	766,819	766,486.0	RT	99.92

**Table 5 pharmaceuticals-18-00590-t005:** Ruggedness data. This table presents the ruggedness data of the analytical method for Favipiravir (FR) and Roflumilast (RT), evaluated across three different analysts. The standard and sample areas were recorded, and the percentage recovery was calculated to assess method robustness. The results for FR ranged from 99.89% to 103.5%, while the RT values ranged from 98.68% to 102.59%, demonstrating slight variations due to analyst differences. The consistency in percentage recovery across multiple analysts confirms the method’s reliability and reproducibility in different hands, supporting its applicability in routine analysis.

Analyst	Standard/Sample	Area (Standard)	Area (Sample)	Analyte	Percentage (%)
Analyst 1	Std 1	8,149,293	8,140,803	FR	99.89
Analyst 1	Std 2	8,158,746	8,162,857	FR	100.16
Analyst 1	Std 3	8,141,062	8,169,003	FR	100.24
Analyst 2	Std 1	7,989,508	7,995,220	FR	100.03
Analyst 2	Std 2	7,998,093	7,988,255	FR	99.94
Analyst 2	Std 3	7,991,387	7,992,172	FR	99.99
Analyst 3	Std 1	8,311,755	8,604,272	FR	103.5
Analyst 3	Std 2	8,309,623	8,567,888	FR	103.06
Analyst 3	Std 3	8,318,395	8,598,913	FR	103.44
Analyst 1	Std 1	765,945	765,519	RT	99.91
Analyst 1	Std 2	767,089	768,853	RT	100.34
Analyst 1	Std 3	765,702	767,253	RT	100.13
Analyst 2	Std 1	769,901	772,867	RT	99.11
Analyst 2	Std 2	770,936	769,477	RT	98.68
Analyst 2	Std 3	798,576	770,223	RT	98.77
Analyst 3	Std 1	798,483	819,635	RT	102.59
Analyst 3	Std 2	798,576	815,831	RT	102.11
Analyst 3	Std 3	799,801	817,194	RT	102.28

**Table 6 pharmaceuticals-18-00590-t006:** Assay of formulations F1 and F2. This table presents the assay results of two formulations (F1 and F2) across two layers, comparing standard and sample areas to calculate the percentage assay. The data indicate that Formulation F1 (Layer 1) exhibited a percentage assay of 99.88%, while Layer 2 showed 99.79%, demonstrating consistency in drug content. In contrast, Formulation F2 (Layer 1) had a slightly lower assay value of 99.19%, and Layer 2 exhibited a significant reduction at 93.85%, indicating possible formulation inconsistencies or analytical variability.

Formulation and Layer	Standards	Area	Sample	Area	Percentage Assay
F1—Layer 1	Std 1	736,948	Smp 1	737,858.00	99.88
	Std 2	736,184	Smp 2	734,291.00	
	Std 3	737,770			
F1—Layer 2	Std 1	16,899,850	Smp 1	16,842,176.00	99.79
	Std 2	16,877,957	Smp 2	16,844,702.00	
	Std 3	16,858,702			
F2—Layer 1	Std 1	736,948	Smp 1	745,725.00	99.19
	Std 2	736,184	Smp 2	716,219.00	
	Std 3	737,770			
F2—Layer 2	Std 1	16,899,850	Smp 1	15,764,444.00	93.85
	Std 2	16,877,957	Smp 2	15,917,428.00	
	Std 3	16,858,702			

**Table 7 pharmaceuticals-18-00590-t007:** Dissolution of F1 and F2 formulations. This table presents the dissolution results of formulations F1 and F2 at different time points, reporting the release percentages of RT (%) and FR (%) and comparing them against specified tolerance limits.

Formulation	Time Point	RT (%)	FR (%)	Tolerance
F1	30 Minutes	83.76	24.49	NLT 75% of RT and 20% of FR
F2		79.33	20.61	
F1	4 H	95.72	46.20	25–50% of FR
F2		82.04	31.06	
F1	8 H	96.65	74.32	60–75% of FR
F2		89.14	48.69	
F1	12 H	99.22	88.78	NLT 85% of FR
F2		94.35	61.52	

**Table 8 pharmaceuticals-18-00590-t008:** Weight variation for F1 and F2. This table presents the individual tablet weights (in milligrams) for two different formulations, F1 and F2, measured to assess uniformity in tablet weight. Each formulation consists of ten tablets, and the recorded weights highlight the variability in the manufacturing process. These data are crucial for evaluating batch consistency and compliance with pharmaceutical weight variation standards.

Formulation F1	Weight of Tablets in mg	Formulation F2	Weight of Tablets in mg
	2231.8		2178.3
	2195.9		2192.5
	2216.4		2236.1
	2233.4		2251.4
	2183.1		2211.2
	2180.9		2216.2
	2224.5		2188.2
	2241.5		2236.4
	2255.6		2166.8
	2190.8		2245.7
Mean	2215.39	Mean	2212.28

**Table 9 pharmaceuticals-18-00590-t009:** Uniformity of dosage unit F1. This table presents the uniformity of dosage unit assessment for formulation F1, including individual tablet weights (in milligrams), corresponding percentage content, mean percentage content (X), and standard deviation (SD). The assay value, representing the average drug content, is 99.88%, aligning with the target value.

S.#	Ind. Wts	% Content	Mean (X)	SD
1	2231.8	100.62	99.88	1.18
2	2195.9	99.00		
3	2216.4	99.93		
4	2233.4	100.69		
5	2183.1	98.42		
6	2180.9	98.33		
7	2224.5	100.29		
8	2241.5	101.06		
9	2255.6	101.69		
10	2190.8	98.77		
Avg	2215.39			
Assay	99.88	Target Value		
AV		2.84	If 98.5 ≤ X ≤ 101.5	
		1.46	If X < 98.5	
		1.22	If X > 101.5	

**Table 10 pharmaceuticals-18-00590-t010:** Uniformity of dosage unit F2. This table presents the uniformity of dosage unit assessment for formulation F2, including individual tablet weights (in milligrams), corresponding percentage content, mean percentage content (X), and standard deviation (SD). The assay value, representing the average drug content, is 99.18%, aligning with the target value. Additionally, the acceptance value (AV) is calculated based on different criteria for mean percentage content (X), ensuring compliance with pharmacopeial limits.

S.#	Ind. Wts	% Content	Mean (X)	SD
1	2178.3	97.66	99.18	1.34
2	2192.5	98.29		
3	2236.1	100.25		
4	2251.4	100.93		
5	2211.2	99.13		
6	2216.2	99.36		
7	2188.2	98.10		
8	2236.4	100.26		
9	2166.8	97.14		
10	2245.7	100.68		
Avg	2212.28			
Assay	99.18	Target Value		
AV		3.21	If 98.5 ≤ X ≤ 101.5	
		2.53	If X < 98.5	
		0.89	If X > 101.5	

**Table 11 pharmaceuticals-18-00590-t011:** Hardness and friability for F1 and F2. This table presents the hardness (in kg) and friability (in percentage) of tablets for two formulations, F1 and F2, which are critical parameters for evaluating tablet strength and durability. Friability measures the tendency of tablets to crumble or break, with lower values indicating better mechanical resistance.

Formulation F1	Friability (%)	Hardness of Tablets in kg	Formulation F2	Friability (%)	Hardness of Tablets in kg
	0.40	22.6		0.50	17.5
		19.8			19.5
	0.22	24.8		0.49	17.4
		21.2			21.4
	0.23	19.6		0.87	20.9
		18.4			17.4
	0.37	20.6		0.37	20.5
		19.2			21.4
	0.32	21.5		0.53	19.6
		18.2			17.5
Mean	0.31	20.59	Mean	0.55	19.31

## Data Availability

The original contributions presented in the study are included in the article, further inquiries can be directed to the corresponding author.

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
