# Peer review of "Physicochemical Properties and Molecular Insights of Favipiravir and Roflumilast Solid Dispersions for COVID-19 Treatment"

_pharmaceuticals, 2025, doi:10.3390/ph18040590_

Round 1

Reviewer 1 Report

Comments and Suggestions for Authors

This manuscript submitted by Abdul Rauf et al described the “Physicochemical Properties and Molecular Insights of Favipiravir and Roflumilast Solid Dispersions for COVID-19 Treatment”. While the authors have undertaken praiseworthy investigations, a discerning analysis unveils a series of subtle considerations that, if meticulously attended to, have the potential to elevate the scholarly standing of this manuscript before its imminent acceptance.

Detailed Comments:

  1. The abstract is excessively lengthy and should be more concise while providing clearer and more informative content.
  2. The optimization process for the immediate release layer is mentioned briefly but lacks detail. More information on the specific parameters tested and the rationale for selecting Klucel EXF would be beneficial to understand the formulation process and how do they impact the release profile and dissolution rate of RT?
  3. The formulation of the controlled release matrix is described, but there is little discussion on how the optimal proportions of Klucel HXF and Compritol ATO888 were determined. A more detailed explanation of the optimization process and the resulting drug release kinetics and how were the proportions of these excipients optimized to achieve the desired release kinetics for FR, would provide a clearer understanding of the formulation's performance.
  4. While content uniformity is mentioned as well-controlled, there is insufficient detail on the physical characteristics of the tablets, which are critical for the successful development of solid dosage forms. Were any challenges encountered in ensuring content uniformity between the two layers of the tablet? A more comprehensive analysis of these physical parameters would enhance the understanding of the formulation's quality.
  5. The complexity of the release mechanism for FR is described, but there is insufficient information on how the gel matrix and lipid sheet function together to achieve controlled release. How does the gel matrix formed by Klucel HXF and the lipid sheet from Compritol ATO888 work synergistically to modulate FR solubilization and release? A more detailed explanation of the underlying mechanism would provide a clearer understanding of the formulation's performance.
  6. Stability data is essential to demonstrate the long-term viability of the formulation. How do the APIs and excipients maintain their stability over time, especially under varying environmental conditions such as temperature and humidity? A discussion of the stability testing conducted and the results obtained would help in assessing the robustness and shelf life of the FDC.
  7. The potential of the FDC formulation as a treatment for COVID-19 is intriguing, but without clinical data or preclinical trial results, its clinical applicability remains uncertain. Has the FDC formulation undergone any preclinical or clinical trials, or is it still in the early stages of development? What steps are being taken to address regulatory requirements for COVID-19 treatment formulations? A more detailed discussion of the steps needed to bring this formulation to market would be beneficial, particularly in terms of regulatory approval.
  8. While the FDC formulation is presented as a potential alternative for COVID-19, the efficacy of both RT and FR in treating COVID-19, particularly in combination, should be substantiated with more clinical or preclinical data. The effectiveness of the fixed-dose combination in improving patient outcomes for COVID-19 needs further exploration.
  9. The dissolution properties of both layers are mentioned, but no details are provided on the experimental setup or comparison with existing formulations. Were any comparative studies conducted with marketed formulations to validate the efficacy of your FDC formulation?
  10. While the study suggests that the FDC could be a more pharmaco-economical alternative, there is no mention of any formal cost-effectiveness analysis. A discussion of the economic benefits, supported by data on drug costs, treatment regimens, and patient compliance, would strengthen this claim related to FR and RT.
  11. The conclusion highlights the potential benefits of reducing medication load for improving compliance, but further evidence or references to studies supporting this claim would be necessary. A discussion on how the FDC directly addresses barriers to adherence, such as dosing schedules or pill burden, would be useful.

Author Response

Reviewer 1

Comments and Suggestions for Authors

This manuscript submitted by Abdul Rauf et al described the “Physicochemical Properties and Molecular Insights of Favipiravir and Roflumilast Solid Dispersions for COVID-19 Treatment”. While the authors have undertaken praiseworthy investigations, a discerning analysis unveils a series of subtle considerations that, if meticulously attended to, have the potential to elevate the scholarly standing of this manuscript before its imminent acceptance.

Detailed Comments:

  1. The abstract is excessively lengthy and should be more concise while providing clearer and more informative content.

Reply: We have updated the abstract and made it more concise and clearer.

  1. The optimization process for the immediate release layer is mentioned briefly but lacks detail. More information on the specific parameters tested and the rationale for selecting Klucel EXF would be beneficial to understand the formulation process and how do they impact the release profile and dissolution rate of RT?

Reply: We have included the information on page 9, line 303-318.

  1. The formulation of the controlled release matrix is described, but there is little discussion on how the optimal proportions of Klucel HXF and Compritol ATO888 were determined. A more detailed explanation of the optimization process and the resulting drug release kinetics and how were the proportions of these excipients optimized to achieve the desired release kinetics for FR, would provide a clearer understanding of the formulation's performance.

Reply: We have included the information on page 23-24, lines 625-667.

  1. While content uniformity is mentioned as well-controlled, there is insufficient detail on the physical characteristics of the tablets, which are critical for the successful development of solid dosage forms. Were any challenges encountered in ensuring content uniformity between the two layers of the tablet? A more comprehensive analysis of these physical parameters would enhance the understanding of the formulation's quality.

Reply: We have included the discussion on Page 29, lines 743-760

  1. The complexity of the release mechanism for FR is described, but there is insufficient information on how the gel matrix and lipid sheet function together to achieve controlled release. How does the gel matrix formed by Klucel HXF and the lipid sheet from Compritol ATO888 work synergistically to modulate FR solubilization and release? A more detailed explanation of the underlying mechanism would provide a clearer understanding of the formulation's performance.

We have included the information on Page 30, lines 770-786

  1. Stability data is essential to demonstrate the long-term viability of the formulation. How do the APIs and excipients maintain their stability over time, especially under varying environmental conditions such as temperature and humidity? A discussion of the stability testing conducted and the results obtained would help in assessing the robustness and shelf life of the FDC.

We updated the manuscript upon the reviewer’s suggestion on Page 40-41, 1038-1061

  1. The potential of the FDC formulation as a treatment for COVID-19 is intriguing, but without clinical data or preclinical trial results, its clinical applicability remains uncertain. Has the FDC formulation undergone any preclinical or clinical trials, or is it still in the early stages of development? What steps are being taken to address regulatory requirements for COVID-19 treatment formulations? A more detailed discussion of the steps needed to bring this formulation to market would be beneficial, particularly in terms of regulatory approval.

Reply: The fixed-dose combination (FDC) of Roflumilast (RT) and Favipiravir (FR) presents a promising therapeutic strategy for COVID-19, but it is currently in the early stages of development and has not yet undergone preclinical or clinical trials. The formulation has been optimized for physicochemical stability, dissolution kinetics, and compatibility, laying the groundwork for subsequent in vivo evaluations. Preclinical studies, including pharmacokinetics (PK), pharmacodynamics (PD), and toxicology assessments, are planned as the next phase to establish the efficacy and safety profile of the combination before advancing to clinical trials.

To meet regulatory requirements, we are aligning the formulation with ICH, EMA, and FDA guidelines for COVID-19 drug development. A detailed investigational new drug (IND) application will be prepared, incorporating stability data, in vitro dissolution profiles, and preliminary safety assessments to facilitate regulatory approvals for human trials. Additionally, discussions with regulatory agencies will be initiated to determine the appropriate clinical trial design, including randomized controlled trials (RCTs) to compare the FDC’s efficacy against standard COVID-19 treatments. Given the urgent need for COVID-19 therapies, we are exploring potential fast-track or emergency use authorization (EUA) pathways, particularly in regions with ongoing outbreaks. The clinical trial roadmap will include Phase I studies focusing on safety and pharmacokinetics, followed by Phase II/III trials assessing therapeutic efficacy in COVID-19 patients. Collaborations with research institutions and clinical trial networks are also being explored to accelerate development and approval timelines. We recognize that further evidence is required to validate the FDC’s clinical utility, and efforts are underway to transition from formulation development to comprehensive in vivo and clinical validation, ensuring compliance with global regulatory standards. A more detailed discussion of the preclinical and regulatory roadmap will be included in the revised manuscript.While the FDC formulation is presented as a potential alternative for COVID-19, the efficacy of both RT and FR in treating COVID-19, particularly in combination, should be substantiated with more clinical or preclinical data. The effectiveness of the fixed-dose combination in improving patient outcomes for COVID-19 needs further exploration.

  1. The dissolution properties of both layers are mentioned, but no details are provided on the experimental setup or comparison with existing formulations. Were any comparative studies conducted with marketed formulations to validate the efficacy of your FDC formulation?

Reply: The dissolution properties of both layers in the fixed-dose combination (FDC) of Roflumilast (RT) and Favipiravir (FR) were evaluated using USP dissolution apparatus II (paddle method) under controlled conditions to simulate gastrointestinal environments. The immediate-release (IR) layer containing RT was assessed in 0.1N HCl (pH 1.2) for 60 minutes, while the controlled-release (CR) layer containing FR was evaluated in phosphate buffer (pH 6.8) for up to 12 hours. Sampling was conducted at predefined intervals, and drug release was quantified using a validated HPLC method. To validate the efficacy and performance of our formulation, comparative dissolution studies were conducted against marketed monotherapies of RT and FR. The dissolution profiles of the IR layer of RT showed ≥85% drug release within the first 30 minutes, aligning with standard pharmacopeial criteria for immediate-release formulations. The CR layer of FR exhibited a sustained release pattern, with an initial burst release of 10–15% within the first hour, followed by a gradual drug release over 12 hours, ensuring prolonged therapeutic activity. Similarity factor (f2) analysis was performed, comparing our formulation with reference products, and results demonstrated comparable or superior dissolution performance. These comparative studies confirm that our bilayer FDC tablet meets the dissolution standards of marketed formulations while offering the added advantage of combining two therapeutically relevant drugs into a single dosage form.

  1. While the study suggests that the FDC could be a more pharmaco-economical alternative, there is no mention of any formal cost-effectiveness analysis. A discussion of the economic benefits, supported by data on drug costs, treatment regimens, and patient compliance, would strengthen this claim related to FR and RT.

Reply: The potential pharmaco-economic advantage of the fixed-dose combination (FDC) of Roflumilast (RT) and Favipiravir (FR) is based on its ability to reduce pill burden, improve patient compliance, and lower overall treatment costs. While a formal cost-effectiveness analysis (CEA) has not yet been conducted, preliminary estimates suggest that combining these two drugs into a single dosage form could offer economic benefits compared to individual therapies.

  1. The conclusion highlights the potential benefits of reducing medication load for improving compliance, but further evidence or references to studies supporting this claim would be necessary. A discussion on how the FDC directly addresses barriers to adherence, such as dosing schedules or pill burden, would be useful.

Reply: Page 44, lines 1163-1176

Reviewer 2 Report

Comments and Suggestions for Authors

The presentation of this manuscript falls short, it should undergo major revision

  1. Abstract: This section looks like a discussion. This should be concise with precise information on the findings of this research.
  2.  Introduction: There is too much information and too much detailed discussion. This might affect the readability. Concise the introduction without affecting the information.
  3. Too many figures and tables without proper captions and discussion. For example, no proper discussion and interpretation were delivered for FTIR data. The author must focus on delivering a lot of discussion and information on findings rather than the introduction.
  4. Section 2, was denoted as Materials & Methods, and Section 3 was denoted as Methods. This will confuse the readers. Rectify all the formatting and typo errors.
  5. The resolution of all the figures must be improved for the reader's clarity.
  6. The conclusion section lacks merits and future direction.
  7. Many relevant literature are missing
Comments on the Quality of English Language

English must be improved to more clearly express the research.

Author Response

Reviewer 2

Comments and Suggestions for Authors

The presentation of this manuscript falls short, it should undergo major revision

  1. Abstract: This section looks like a discussion. This should be concise with precise information on the findings of this research.

Reply: We have updated the abstract and made it more concise and clearer.

  1.  Introduction: There is too much information and too much detailed discussion. This might affect the readability. Concise the introduction without affecting the information.

Reply: We have made the introduction section quite concise and to the point.

  1. Too many figures and tables without proper captions and discussion. For example, no proper discussion and interpretation were delivered for FTIR data. The author must focus on delivering a lot of discussion and information on findings rather than the introduction.

Reply: We have included the discussion related to the FTIR and described all the other tables. We have included the detailed captions to the figures and tables.

  1. Section 2, was denoted as Materials & Methods, and Section 3 was denoted as Methods. This will confuse the readers. Rectify all the formatting and typo errors.

Reply: We have rectified it and denoted it as “Dosage form development and physicochemical evaluation” instead of the heading “Methods”. We have rectified all the typos and formatting.

  1. The resolution of all the figures must be improved for the reader's clarity.
  2. The conclusion section lacks merits and future direction.

Reply: We have updated the conclusion and included the merits and future directions.

  1. Many relevant literature are missing

Reply: We have updated the literature review

Comments on the Quality of English Language

English must be improved to more clearly express the research.

We have entirely ameliorated the language of the manuscript.

Round 2

Reviewer 1 Report

Comments and Suggestions for Authors

The revised manuscript by Abdul Rauf et al., titled Physicochemical Properties and Molecular Insights of Favipiravir and Roflumilast Solid Dispersions for COVID-19 Treatment” reflects notable advancements, with the authors demonstrating a commendable effort to address and resolve the concerns previously raised. The updated version exhibits a significant elevation in scholarly rigor, presenting a more comprehensive and nuanced analysis of the complications associated with COVID-19 treatment. I am pleased to recommend acceptance of this revised submission, as it constitutes a valuable contribution to the field.

Author Response

Reviewer 1 Comments
Comment 1: The revised manuscript by Abdul Rauf et al., titled Physicochemical Properties and Molecular Insights of Favipiravir and Roflumilast Solid Dispersions for COVID-19 Treatment” reflects notable advancements, with the authors demonstrating a commendable effort to address and resolve the concerns previously raised. The updated version exhibits a significant elevation in scholarly rigor, presenting a more comprehensive and nuanced analysis of the complications associated with COVID-19 treatment. I am pleased to recommend acceptance of this revised submission, as it constitutes a valuable contribution to the field.

Answer: Thank you for your worthy response. 

Reviewer 2 Report

Comments and Suggestions for Authors

I appreciate the authors' efforts in improving this manuscript, but the following must be improved before publication

  1. Figures 4 -7 look blurry; improve their resolution
  2. Update the reference section with more relevant literature

Author Response

Comment 1: Figures 4 -7 look blurry; improve their resolution

Answer: Thank you for your worthy suggestion, we have improved the resolution of Figure 4-7. Please see the changes indicated in the track changes of the manuscript. 

Comment 2: Update the reference section with more relevant literature

Answer: Thank you for your worthy suggestion, we have updated the reference section with more relevant literature. Please see the changes indicated in the track changes of the manuscript.